# Genotoxic stress-triggered β-catenin/JDP2/ PRMT5 complex facilitates reestablishing glutathione homeostasis

Lixue Cao[1,2,6], Geyan Wu[1,3,6], Jinrong Zhu[1,2], Zhanyao Tan[2], Dongni Shi[3], Xingui Wu[2], Miaoling Tang[3], Ziwen Li[2], Yameng Hu[2], Shuxia Zhang[2], Ruyuan Yu[2], Shuang Mo[2], Jueheng Wu[4], Erwei Song[5], Mengfeng Li [4], Libing Song[3] & Jun Li[1,2]

The mechanisms underlying how cells subjected to genotoxic stress reestablish reduction-oxidation (redox) homeostasis to scavenge genotoxic stress-induced reactive oxygen species (ROS), which maintains the physiological function of cellular processes and cell survival, remain unclear. Herein, we report that, via a TCF-independent mechanism, genotoxic stress induces the enrichment of β-catenin in chromatin, where it forms a complex with ATM phosphorylated-JDP2 and PRMT5. This elicits histone H3R2me1/H3R2me2s-induced transcriptional activation by the recruitment of the WDR5/MLL methyltransferase complexes and concomitant H3K4 methylation at the promoters of multiple genes in GSH-metabolic cascade. Treatment with OICR-9429, a small-molecule antagonist of the WDR5-MLL interaction, inhibits the β-catenin/JDP2/PRMT5 complex-reestablished GSH metabolism, leading to a lethal increase in the already-elevated levels of ROS in the genotoxic-agent treated cancer cells. Therefore, our results unveil a plausible role for β-catenin in reestablishing redox homeostasis upon genotoxic stress and shed light on the mechanisms of inducible chemotherapy resistance in cancer.

[1] Key Laboratory of Liver Disease of Guangdong Province, The Third Affiliated Hospital, Sun Yat-sen University, Guangzhou, China. [2] Department of Biochemistry, Zhongshan school of medicine, Sun Yat-sen University, Guangzhou, China. [3] State Key Laboratory of Oncology in South China, Collaborative Innovation Center for Cancer Medicine, Sun Yat-sen University Cancer Center, Guangzhou, China. [4] Department of Microbiology, Zhongshan School of Medicine, Sun Yat-sen University, Guangzhou, China. [5] Department of Breast Surgery, Sun Yat-Sen Memorial Hospital, Sun Yat-Sen University, Guangzhou, China. [6] These authors contributed equally: Lixue Cao, Geyan Wu. Correspondence and requests for materials should be addressed to J.L. (email: lijun37@mail.sysu.edu.cn)

Reactive oxygen species (ROS) are composed of free radicals with unpaired electron and non-radical oxygen species containing oxygen, such as superoxide ($O_2^{-\cdot}$), hydrogen peroxide ($H_2O_2$), singlet oxygen ($^1O_2$), and the hydroxyl radical (HO·)[1–3]. Acting as signaling molecules, ROS are essential for the efficient and proper execution of a large number of cellular processes, such as regulation of intracellular signal transduction and gene expression patterns[1–3]. However, excessive or prolonged ROS generation results in considerable damage to cellular constituents, various diseased conditions, and the process of ageing[4,5]. On the other hand, intracellular thiols, such as glutathione (GSH), cysteine (Cys), and homocysteine (Hcy), play crucial roles in defense against oxidative stress and scavenging of ROS, resulted in maintaining biological redox homeostasis[6,7]. Hence, maintaining the redox homeostasis or reestablishing the redox balance in response to stress stimuli is fundamental for physiological cellular function and survival. As the most abundant endogenous low-molecular-weight redox molecule within mammalian cells, glutathione (GSH) plays pleiotropic roles in preventing damage induced by either external or intracellular stimuli. GSH either functions as an antioxidant to scavenge ROS directly or serves as an electron donor for other redox systems, such as glutaredoxin (Grx) and glutathione peroxidase (GPX), to scavenge peroxide-related products. Meanwhile, GSH could also represent a storage sink for cysteine, which displays cellular toxicity when present at high concentrations, and exert detoxification effects by conjugation with and exporting toxicants and xenobiotic compounds out of cells[8–10]. Thus, maintaining optimal intracellular GSH levels are crucial for cellular homeostasis and organismal fitness.

Under physiological conditions, intracellular levels of GSH are mostly maintained by de novo synthesis from the precursor amino acids cysteine, glutamate, and glycine, which process is mediated by multiple components in the GSH-metabolic cascade, including the cystine–glutamate transporter SLC7A11 and the GSH rate-limiting enzymatic γ-glutamylcysteine ligase (GCL) and GSH synthetase (GSS)[11,12]. Meanwhile, the salvage pathways, such as γ-glutamyl transferase (GGT) and thioredoxin/glutaredoxin (TRX/GRX), have been also demonstrated to contribute to GSH homeostasis[13–15]. For instance, γ-GGT could enhance cellular GSH synthesis by increasing the availability of component amino acids, and TRX/GRX could regulate cellular GSH homeostasis by reduction of oxidized forms, such as glutathione disulfide (GSSG) and glutathione mixed disulfide with protein thiols (GS-R), back to the reduced form of GSH[13–15]. By contrast, under conditions of intensive external insults, such as genotoxic stress, high levels of different types of intracellular ROS were induced via several mechanisms, which resulted in prominent depletion of cellular stores of the reduced form of GSH[16–18]. Therefore, restoring GSH levels in cells subjected to genotoxic stress is crucial to maintain physiological cellular function and survival. Interestingly, the genotoxic stress-mediated reduction of GSH level could speedily recovery back and even further elevated after a few hours[17,18]. However, the mechanisms underlying the genotoxic stress-treated cells reestablished GSH homeostasis remain unclear.

In this study, we report that genotoxic stress activates TCF-independent β-catenin signaling that contributes to reestablishing GSH metabolism and the rapid reduction in genotoxic stress-induced ROS. We demonstrate that β-catenin forms a complex with Jun Dimerization Protein 2 (JDP2) and arginine methyltransferase 5 (PRMT5) that elicit histone H3R2me1/H3R2me2s-induced transcriptional activation via a TCF-independent mechanism, by recruitment of WD repeat domain 5 (WDR5)/myeloid/lymphoid or mixed-lineage leukemia protein (MLL) methyltransferase complexes at the promoters of multiple genes

in the GSH-metabolic cascade. Taken together, our results unveil a plausible role for β-catenin in reestablishing redox homeostasis upon genotoxic stress.

## Results

### β-catenin regulates GSH metabolism upon genotoxic stress.
The β-catenin signaling pathway plays a central role in various cellular processes via TCF-dependent and TCF-independent mechanisms[19–21]. Interestingly, we observed that in response to genotoxic stresses induced by camptothecin (CPT), irradiation (IR), or cisplatin (CDDP), 293FT, OVCAR3, MCF-7, and A549 cells exhibited rapid enrichment of β-catenin in their chromatin, in less than 15 min, but displayed decreased β-catenin/TCF4 complex formation and transcriptional activity (Fig. 1a–d; Supplementary Fig. 1a–d). Interestingly, we did not find that genotoxic stresses induced the β-catenin/FOXO3 interaction, which has been reported to play a vital role in regulation of oxidative stress signaling[22,23] (Supplementary Fig. 1d). Therefore, these results suggested that genotoxic stress-induced β-catenin signaling might be activated via a TCF- or FOXO-independent mechanism.

To further investigate the biological role of genotoxic stress-activated-β-catenin signaling, β-catenin chromatin immunoprecipitation (ChIP-seq) and RNA sequencing (RNA-seq) assays were conducted in CPT-treated 293FT cells. Analysis of pooled ChIP-seq data using two replicate data sets, correlated significantly with each other ($P < 1.0 \times 10^{-10}$, $r = 0.85$ by Spearman's chi-squared test), showed 57.4 million total reads and 55.6 million mapped reads, which β-catenin signals was associated with 20,521 peaks, including 6925 peaks in the promoter, 4265 peaks in intergenic, 6821 peaks in intron, 1112 peaks in the exon, 688 peaks in 5′ UTR, 125 peaks in 3′ UTR, 365 peaks in TTS in CPT-treated 293FT cells (PRJNA543097) (Supplementary Table 1). The RNA-seq data sets analysis showed that comparing the gene expression profiles of β-catenin siRNA with scramble transfectants, a total of 292 downregulated genes (fold change ≥ 2.0-fold) were detected in both β-catenin-silenced 293T cells treated by CPT (PRJNA543096). Interestingly, we found that the genes with GO terms, "Cell redox homeostasis", "Glutathione metabolic process", "Negative regulation of response to ROS", and "Negative regulation of ROS metabolic process" were significantly enriched in both the RNA- and ChIP-seq profiles (Fig. 2a, b; Supplementary Fig. 2a), suggesting that genotoxic stress-activated-β-catenin signaling might be involved in the GSH metabolic process. Consistent with this hypothesis, we found that β-catenin in the genotoxic-stressed cells was associated with certain gene promoters and contributed to the transcriptional upregulation of SLC7A11, GCLM, and GSS, which encode key factors in the GSH-metabolic cascade (Fig. 2c; Supplementary Fig. 2b–d). Importantly, silencing β-catenin not only abolished the genotoxic stress-induced upregulation of these GSH-metabolic genes but also abrogated the rapid restoration of intracellular GSH production, which resulted in sustained ROS levels and increased numbers of 8-Oxo-2′-deoxyguanosine (8OHdG)-positive cells (Fig. 2d–f; Supplementary Fig. 2d–h). However, we did not observe that mRNA expressions of γ-GGT, TRX, and GRX genes, the key regulator of GSH in salvage pathways[13–15], were significantly altered in genotoxic stress-treated cells via RNA-seq analysis. Taken together, our results suggested that genotoxic stress-activated β-catenin signaling facilitates the restoration of GSH metabolism via de novo GSH synthesis.

### JDP2 is involved in GSH metabolism upon genotoxic stress.
Consistently, silencing TCF/LEF factors, including TCF1(TCF7),

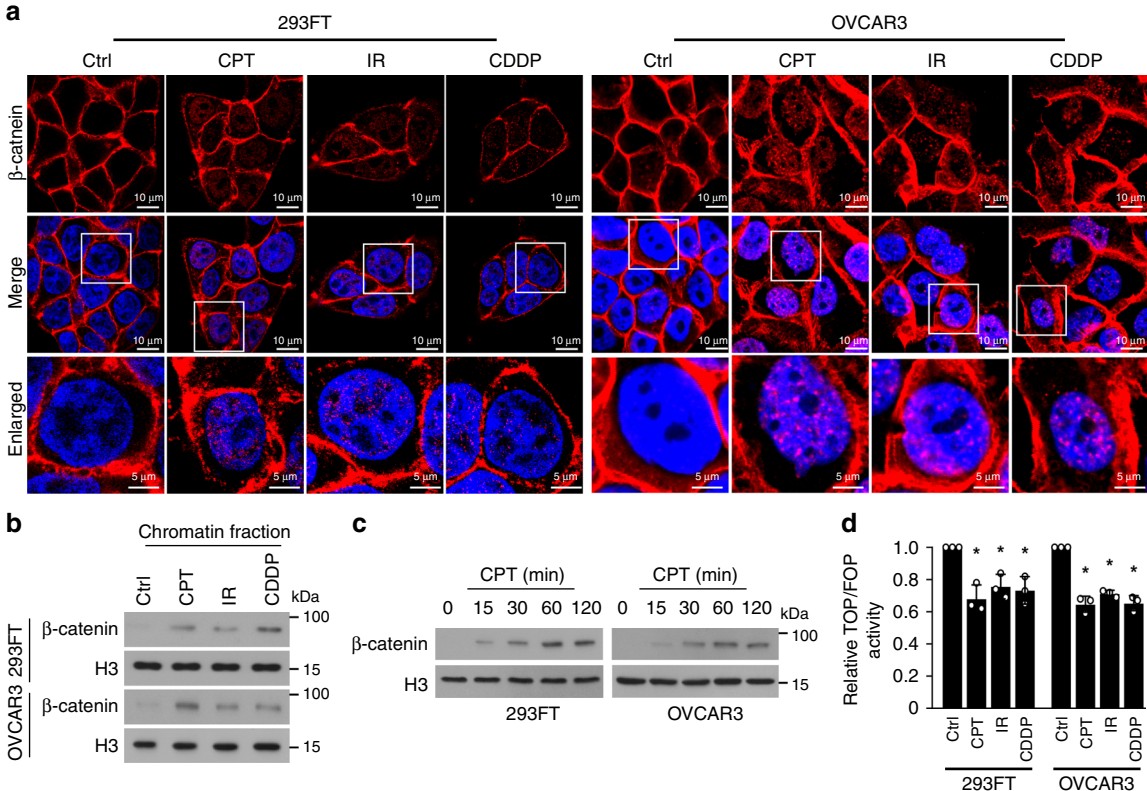

**Fig. 1** Genotoxic stress-induced β-catenin signaling is activated via a TCF- or FOXO-independent mechanism. **a** Representative images of the subcellular localization of β-catenin in the indicated cells treated with CPT (10 μM, 1 h), IR (10 Gy), and CDDP (10 μM, 1 h), as analyzed by immunofluorescence staining. Scale bar, upper and middle panel: 10 μm; lower panel: 5 μm. **b** IB analysis of expression of β-catenin expression in the chromatin fraction extracted from the indicated cells treated with CPT (10 μM, 1 h), IR (10 Gy), and CDDP (10 μM, 1 h). Histone 3 served as the loading control. **c** IB analysis of β-catenin expression in the chromatin fraction extracted from the indicated cells treated with CPT (10 μM) for 0, 15, 30, 60, and 120 min. Histone 3 served as the loading control. **d** Relative TOP flash or FOP flash luciferase reporter activity was analyzed in the indicated cells treated with CPT (10 μM, 1 h), IR (10 Gy), and CDDP (10 μM, 1 h). Each error bar represents the mean ± SD of three independent experiments. *$P < 0.05$. Student's two-tailed $t$ test. Source data of Fig. 1d are provided as a Source Data file

LEF1, TCF3(TCF7L1), and TCF4 (TCF7L2), had no effects on the expression of GSH-metabolic genes and the enrichment of β-catenin on the promoters of these GSH-metabolic genes in genotoxic agent-treated cells (Supplementary Fig. 3a–c), which suggested that other transcription factor(s) might be involved in genotoxic stress-activated β-catenin signaling-mediated GSH metabolism. We then performed an immunoprecipitation (IP) assay using the chromatin fractions derived in CPT-treated β-catenin-transduced 293FT cells and excised five remarkable different bands precipitated by β-catenin antibody for mass spectrometry (MS) analysis. As shown in Supplementary Data 1–2, there were 57 proteins identified to be potent β-catenin-binding proteins. Interestingly, among these binding proteins, α-catenin and poly (ADP-ribose) polymerase 1 (PARP1), have been previously reported to be β-catenin-interacting protein in the genotoxic stress-treated cells[21,24]. The proteins with more than five peptides identified by MS, including SMARCA4, PARP1, α-catenin, PRMT5, FOXO3, TCF4, HNRNPA2B1, and JDP2, were selected for further examination (Fig. 3a). As shown in Fig. 3b, c and Supplementary Fig. 4a–d, silencing JDP2 and PRMT5 in genotoxic stress-treated cells significantly decreased the expression of GSH-related genes, but only silencing JDP2 reduced the enrichment of β-catenin on the promoters of these genes. These results suggested that JDP2 might be a transcriptional factor that contributes to the association of β-catenin with promoters of GSH-related genes in cells subjected to genotoxic stress. Furthermore, we found that overexpressing JDP2 dramatically

increased, but silencing JDP2 decreased, the expression of GSH-metabolic genes and GSH level (Fig. 3d, e), and that genotoxic stress-induced ROS production was also rapidly decreased in JDP2-transduced cells but was prolonged in JDP2-silenced cells (Fig. 3e). These results demonstrate a crucial role of JDP2 in GSH metabolism upon genotoxic stress.

**β-catenin interacts with JDP2 upon genotoxic stress**. Co-IP assays revealed that β-catenin formed a complex with JDP2 and PRMT5 only in CPT-treated cells, but not in untreated cells (Fig. 3f; Supplementary Fig. 5a, b), suggesting that the β-catenin/JDP2/PRMT5 complex only formed in cells subjected to genotoxic stress. However, silencing JDP2 almost entirely abrogated the β-catenin/PRMT5 interaction, while downregulating β-catenin did not reduce the JDP2/PRMT5 interaction, and ablating PRMT5 had no obvious impact on the JDP2/β-catenin association (Fig. 3g), which indicated that JDP2 was essential for the formation of the β-catenin/JDP2/PRMT5 complex. Far-western blotting and stochastic optical reconstruction microscopy (STORM) analyses further confirmed the direct interaction of β-catenin with JDP2 in CPT-treated cells (Fig. 3h, i). Moreover, co-IP assays using serially truncated β-catenin fragments revealed that JDP2 interacted with the 3rd–6th armadillo repeats of β-catenin (Fig. 3j), which is the TCF4-interacting region of β-catenin, indicating that upon genotoxic stress, JDP2 competed with TCF4 for β-catenin interaction to regulate downstream

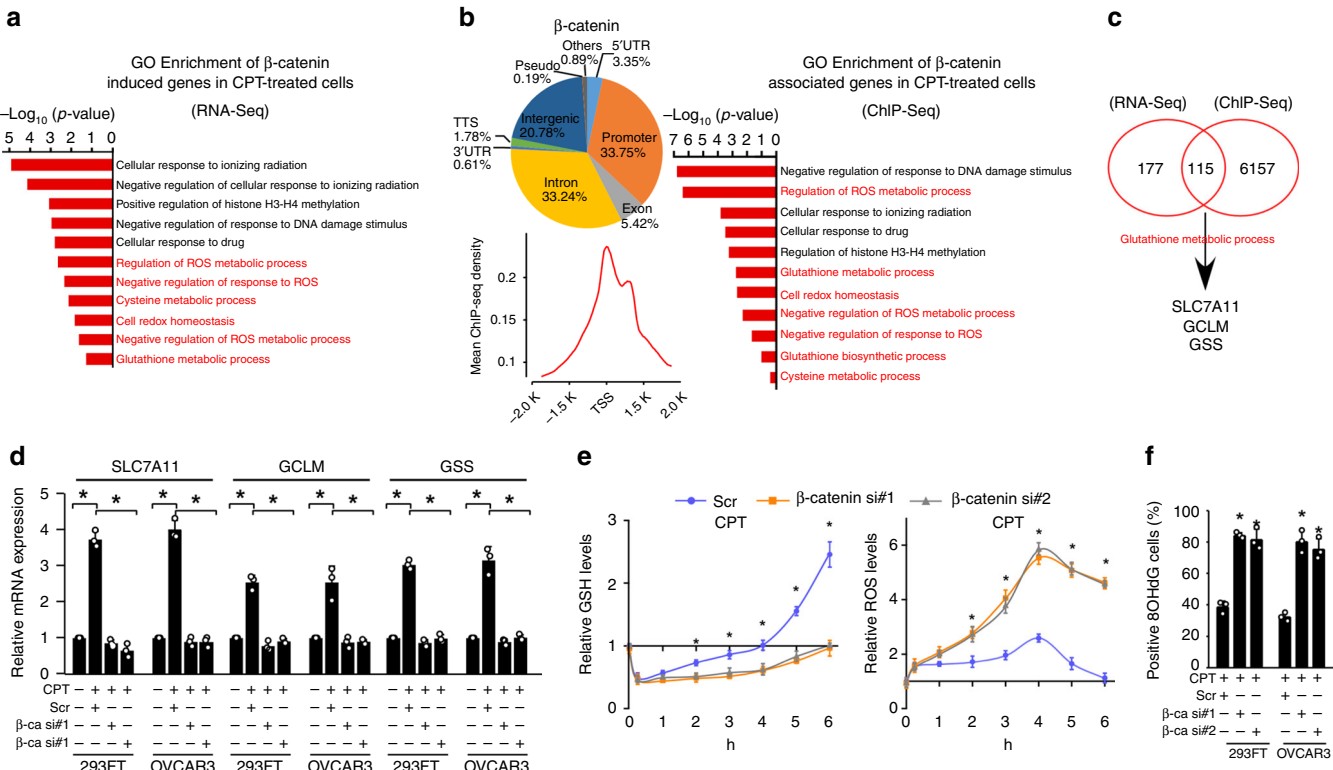

**Fig. 2** β-catenin contributes to genotoxic stress-activated glutathione metabolic processes. **a, b** GO enrichment analysis of β-catenin-regulated transcripts identified using RNA-seq (**a**, PRJNA543096) or ChIP-seq (**b**, PRJNA543097) profiling in CPT (10 μM, 1 h)-treated 293FT cells. The x-axis shows the enrichment scores as calculated by the −log10 (p-value). **b** (left): occupation and average ChIP-seq peak calling around the β-catenin peaks' center. **c** Analysis of RNA-seq (PRJNA543096) and ChIP-seq (PRJNA543097) profiles indicated that β-catenin targeted the promoters and transcriptional upregulated the expression of *SLC7A11*, *GCLM*, and *GSS* in CPT (10 μM, 1 h)-treated cells. **d** Relative expression of *SLC7A11*, *GCLM*, and *GSS* in CPT (10 μM, 1 h)-treated cells as quantified by qRT-PCR analysis. + : treatment, − : untreatment. **e** Relative expression of GSH (left) and ROS (right) were examined in scramble or β-catenin siRNA(s) transfected-cells treated with CPT (10 μM) at the indicated time. **f** The percentage of 8OHdG-positive cells in CPT (10 μM, 4 h)-treated 293FT and OVCAR3 cells analyzed using an 8OHdG staining assay. + : treatment, − : untreatment. Each error bar in panels **d** and **f** represents the mean ± SD of three independent experiments. *P < 0.05. Student's two-tailed t test. Source data of Fig. 2d–f are provided as a Source Data file

GSH-metabolic genes. Indeed, overexpressing JDP2 in CPT-treated cells dramatically decreased the formation and transcriptional activity of the TCF4/β-catenin complex (Supplementary Fig. 5c, d). Meanwhile, genotoxic stress dramatically increased the enrichment of JDP2 and β-catenin, but not TCF4 and FOXO3, on the *SLC7A11* promoter (Supplementary Fig. 5e). Therefore, our results demonstrated that genotoxic stress-activated β-catenin signaling-induced GSH metabolism depends on JDP2, but is independent of TCFs.

**β-catenin promotes DNA-binding activity of JDP2.** Upregulation of JDP2 further enhanced, while downregulation of JDP2 abrogated, β-catenin-induced GSH production and ROS diminution in genotoxic agent-treated cells (Fig. 4a). Meanwhile, silencing of β-catenin also drastically abolished the genotoxic stress-induced enrichment of JDP2 on the promoters of GSH-metabolic genes (Fig. 4b). Therefore, we hypothesized that the binding of β-catenin to JDP2 might promote JDP2 to target the downstream gene promoters. Previous studies showed that JDP2-induced repression of downstream targets was caused by inhibition of p300-mediated acetylation of core histones via direct binding to histone H3/H4 through its histone-binding domain and reconstituting nucleosomes[25,26]. Interestingly, our reciprocal co-IP assays using serially truncated JDP2 fragments showed that β-catenin interacted with the histone-binding domain of JDP2 (Fig. 4c), suggesting that the binding of β-catenin to JDP2 inhibited the JDP2/histones interaction. This hypothesis was confirmed using in vivo and in vitro binding assays, which showed that genotoxic stress resulted in decreased association of JDP2 with histone H3/H4, which was further reduced in β-catenin-overexpressing cells, but abolished in β-catenin-silenced cells (Fig. 4d–f; Supplementary Fig. 5f). Moreover, electrophoretic mobility shift assay (EMSA) analysis revealed that overexpressing β-catenin dramatically enhanced, but silencing β-catenin decreased, the DNA-binding activity of JDP2 (Fig. 4g). Taken together, our results demonstrated that the β-catenin-induced JDP2 DNA-binding activity is attributed to inhibiting the association of JDP2 with histones.

**PRMT5 promotes β-catenin/JDP2-activated GSH metabolism.** Although PRMT5 has no effect on the JDP2/β-catenin interaction or the enrichment of JDP2 on the promoters of GSH-related genes (Figs. 3g, 5a), silencing PRMT5 significantly abolished the inductive effect of JDP2 on GSH-metabolic gene expression and delayed GSH production, resulting in prolonged elevated levels of ROS in genotoxic agent-treated cells (Fig. 5b–e; Supplementary Fig. 6a, b). These results suggested that PRMT5 is also involved in β-catenin/JDP2-induced GSH metabolism. This hypothesis was further confirmed by the ChIP assay that silencing either β-catenin or JDP2 could inhibit the enrichment of PRMT5 on the *SLC7A11*, *GCLM*, and *GSS* promoter (Supplementary Fig. 6c). Moreover, co-IP assays using serially truncated JDP2 fragments and PRMT5 demonstrated that PRMT5 interacts with the leucine zipper domain (LZD) of JDP2 (Fig. 5f), which is also the

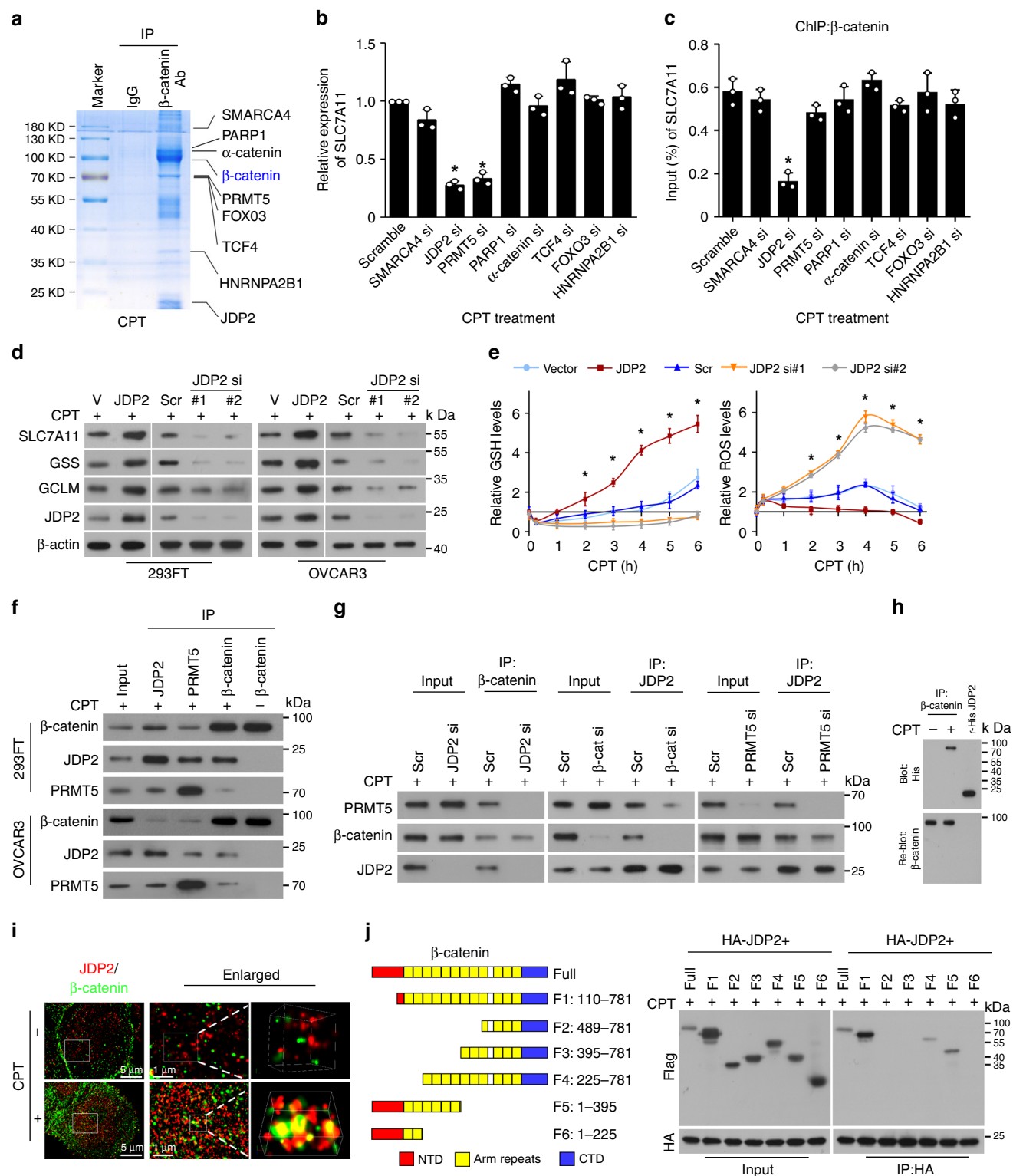

interacting region of transcription factor AP-1[26,27]. Consistently, the binding affinity of JDP2 to ATF3, HDAC3, and c-Jun was dramatically decreased in CPT-treated cells (Fig. 5g). Therefore, these results suggested that genotoxic stress might abolish JDP2-mediated AP-1 transcriptional repression.

**ATM induces JDP2 phosphorylation.** Genotoxic stress markedly promoted the formation of the JDP2/PRMT5 complex (Fig. 5h), which prompted us to examine whether genotoxic stress-induced

DNA damage signaling was involved. Interestingly, we found that inhibition of ATM kinase activity, a major DNA damage-activated kinase, drastically reduced the JDP2/PRMT5 interaction, whereas treated with a phosphatase or overexpressing the JDP2/T116A mutant, a mutant with ATM consensus phosphorylation site, severely impaired the genotoxic stress-induced phosphorylation of JDP2 (Fig. 5i, j), suggesting that genotoxic stress-activated ATM phosphorylated-JDP2 at the T116 site. Consistently, PRMT5 could not associate with the JDP2/T116A

**Fig. 3** JDP2 is essential for β-catenin-induced GSH metabolism upon genotoxic stress. **a–c** IP assays were performed in the chromatin fraction extracted from CPT (10 μM, 1 h)-treated β-catenin-transduced 293FT cells using anti-β-catenin antibody or anti-IgG antibody, followed by mass spectrometry analysis. **b** Relative expression of *SLC7A11* in the indicated siRNA-transfected cells treated with CPT (10 μM, 1 h) as quantified by qRT-PCR analysis. **c** ChIP assay analysis of the enrichment of β-catenin on the *SLC7A11* promoter in the indicated siRNA-transfected cells treated with CPT (10 μM, 1 h). **d** IB analysis of the expression of the indicated protein in CPT (10 μM, 4 h)-treated cells transfected with vector, JDP2, scramble, or JDP2-siRNA(s). β-actin served as the loading control. **e** Relative expression of GSH (left) and ROS (right) were examined in CPT (10 μM)-treated cells at the indicated times. **f** IP assays revealing that β-catenin formed a complex with JDP2 and PRMT5 in CPT (10 μM, 1 h)-treated cells. **g** IP assays were performed in CPT (10 μM)-treated cells and showed that silencing of JDP2 expression reduced the interaction between β-catenin and PRMT5 (left), whereas downregulating β-catenin had no impact on the JDP2/PRMT5 interaction (middle) and downregulating PRMT5 had no impact on the JDP2/β-catenin interaction (right). **h** Far-western blotting analysis was performed using anti-β-catenin antibody-immunoprecipitated proteins and detected using anti-His antibody and then reblotted with anti-β-catenin antibody. Recombinant JDP2 served as the control. **i** The interaction of β-catenin and JDP2 was examined in the control and CPT (10 μM, 1 h)-treated cells using STORM captured in a wide shot (left; scale bar, 5 μm), further zoomed-in (middle; scale bar, 1 μm), and 3D-rendered (right). **j** Schematic illustration of the wild-type and truncated β-catenin protein (left) and co-IP assays were performed using anti-JDP2 antibody in the CPT (10 μM, 1 h)-treated cells transfected with truncated β-catenin fragments (right). Each error bar in panels **b**, **c**, and **e** represents the mean ± SD of three independent experiments. *P < 0.05. Student's two-tailed *t* test. Source data of Fig. 3b, c and 3e are provided as a Source Data file

mutant in the genotoxic agent-treated cells, but showed higher binding affinity with phosphomimetic JDP2/T116D mutant even in untreated cells (Fig. 5k). In addition, we observed that the phosphomimetic JDP2/T116D mutant drastically reduced the binding affinity of JDP2 to AP-1 and HDAC (Supplementary Fig. 6d). These results indicated that ATM-mediated JDP2 phosphorylation is required for the JDP2/PRMT5 interaction. Moreover, co-IP assays revealed that genotoxic stress induced the interaction of ATM with JDP2 via the LZD of JDP2 (Fig. 5l, m). Therefore, our results suggested that genotoxic stress induces JDP2 phosphorylation and JDP2/PRMT5 interaction.

**PRMT5 epigenetically enhances GSH metabolism**. PRMT5 plays roles in transcriptional activation or repression via diverse histone methylation modifications[28–30]. Prominently, genotoxic stress-induced significantly increased levels of H3R2me1 and H3R2me2s on the *SLC7A11* promoter, which was markedly abolished by β-catenin-, JDP2-, and PRMT5-silencing (Fig. 6a, b). Furthermore, we found that inhibiting PRMT5 activity using the PRMT5 inhibitor GSK591 also significantly reduced the levels of H3R2me1 and H3R2me2s on the *SLC7A11* promoter in CPT-treated cells (Fig. 6c). These results suggested that PRMT5-mediated histone methylation contributes to β-catenin/JDP2-activated GSH metabolism. Histone H3R2me1 or H3R2me2s modification-mediated transcriptional activation has been reported to act via recruiting WDR5/MLL methyltransferase complexes, which induced histone H3K4me3, a mark that is recognized by the RNA polymerase II transcription complex on the promoters of target genes[28]. Consistently, we observed that the genotoxic stress-triggered β-catenin/JDP2/PRMT5 complex significantly increased the levels of WDR5 and MLL1~3, but not MLL4 and MLL5, at the *SLC7A11* promoter (Fig. 6d, e). However, silencing WDR5 or treatment with an antagonist of the WDR5/MLL interaction via small-molecule OICR-9429 not only significantly reduced the expression of GSH-metabolic genes but also decreased the intracellular cysteine and GSH levels, which resulted in a lethal elevation of already high levels of ROS and increased the proportion of apoptotic cells in the genotoxic agent-treated cells (Fig. 6f–i). Therefore, these results suggested that PRMT5 contributes epigenetically to JDP2/β-catenin-activated GSH metabolism upon genotoxic stress.

**JDP2 level correlates with poorer survival of cancer patient**. Online Kaplan–Meier plotter analysis revealed that patients with high JDP2 expression in ovarian, lung, gastric, or breast cancer had significantly shorter progression-free survival and shorter overall survival than patients with low JDP2 expression (Fig. 7a, b),

suggesting that a higher JDP2 level correlated with cancer relapse and poorer patient outcome. Ovarian cancer is one of the most common recurrent tumors. Therefore, we further examined the correlation of JDP2 levels with clinicopathological characteristics in 146 clinical ovarian cancer samples. As shown in Fig. 7c–f and Supplementary Tables 2–3, statistical analysis revealed that JDP2 levels were significantly correlated with CDDP resistance ($P <$ 0.001; $r = 0.37$), relapse ($P = 0.002$; $r = 0.25$), FIGO stage ($P = 0.009$; $r = 0.21$), SLC7A11 expression ($P = 0.008$; $r = 0.22$) and GSH level ($P = 0.007$; $r = 0.482$), but was associated with shorter overall/relapse-free survival in patients with ovarian cancer treated with platinum-based therapy (both $P < 0.05$). Moreover, the positive correlation between JDP2 expression and genotoxic stress was further confirmed by gene set enrichment analysis (GSEA), in which JDP2 expression correlated strongly with gene signatures for cisplatin resistance (Fig. 7g). These results indicated the JDP2 contributes to therapeutic resistance of cancer.

**JDP2 confers resistance to genotoxic stress on cancer cells**. To further determine the effect of JDP2 on cancer therapeutic resistance, the gain- or loss-of-function of JDP2 were tested in ovarian and lung cancer cell models (Fig. 8a). Apoptosis and clonogenic survival assays showed that upregulation of JDP2 significantly increased the resistance of OVCAR3 and A549 cells to CPT treatment, accompanied by decreased levels of cleaved-PARP1 and -Caspase 3 (Fig. 8b–d). In contrast, JDP2-silenced cells upon CPT treatment exhibited a significantly higher apoptotic rate and increased levels of activated-PARP1 and -Caspase 3, but showed reduced colony formation (Fig. 8b–d). These results demonstrated that JDP2 contributes to the resistance of cancer cells to genotoxic treatment in vitro.

Furthermore, the effect of JDP2 dysregulation on genotoxic stress was examined using an in vivo intraperitoneal ovarian cancer mouse model treated with Topotecan, a common chemotherapeutic drug widely used to treat ovarian, lung, and other cancers. As shown in Fig. 9a–d, the tumors formed by JDP2-transduced cells upon Topotecan chemotherapy sustained a higher growth rate, as indicated by fewer TUNEL$^+$-cells, and exhibited higher GSH concentrations, but lower ROS levels, resulting in the shorter survival of tumor-bearing mice. By contrast, silencing JDP2 via a short interfering RNA (siRNA), incorporated into dioleoylphosphatidylcholine (DOPC) nanoliposomes, dramatically enhanced the anti-tumor effect of Topotecan, resulting in a lower growth rate of tumor and GSH levels, but higher ROS levels and apoptosis-positive cells in the tumors (Fig. 9a–d). Taken together, these results further supported the notion that JDP2 contributes to resistance to

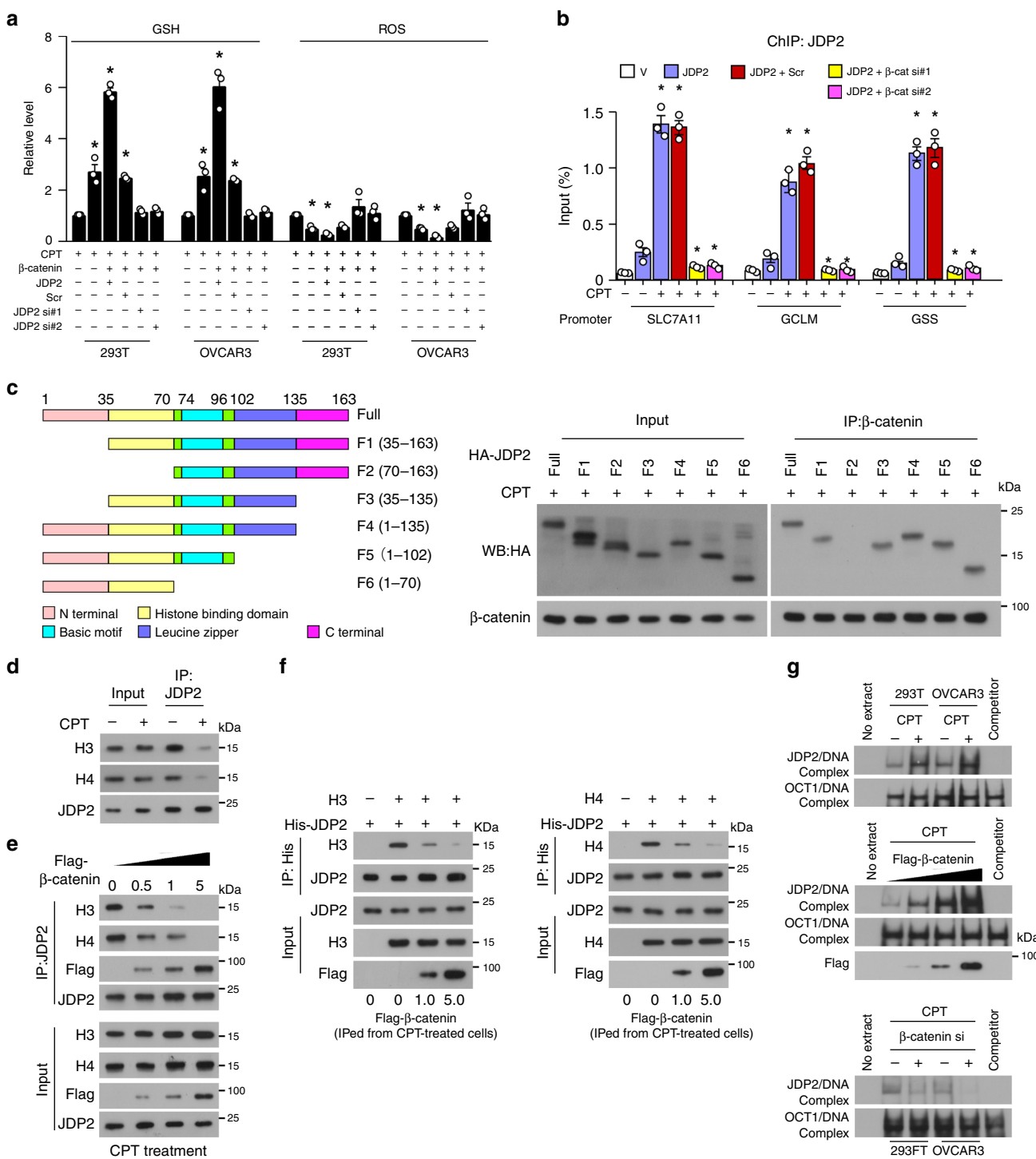

**Fig. 4** β-catenin promotes the DNA-binding activity of JDP2 via inhibition of the JDP2/histone interaction. **a** Relative expression of GSH (left) and ROS (right) were examined in the indicated cells treated with CPT (10 μM, 4 h). +: treatment, −: untreatment. **b** ChIP assay analysis of the enrichment of JDP2 on the promoters of *SLC7A11*, *GCLM*, and *GSS* in the indicated cells treated with or without CPT (10 μM, 1 h). **c** Schematic illustration of the wild-type and truncated JDP2 (left); co-IP assays were performed using anti-β-catenin antibody in the CPT (10 μM, 1 h)-treated cells transfected with truncated JDP2 fragments (right). **d** IP assays were performed using anti-JDP2 antibody in the indicated cells treated with or without CPT (10 μM, 1 h) and IB analysis of expression of JDP2, Histone 3, and Histone 4. **e** IP assays were performed using anti-JDP2 antibody in CPT (10 μM, 1 h)-treated cells transfected with 0, 0.5, 1.0, and 5.0 μg of a Flag-tagged β-catenin plasmid and IB analysis of expression of JDP2, Flag-tagged β-catenin, Histone 3, and Histone 4. **f** In vitro binding assays were performed using anti-His antibody in the reactions mixed with recombinant His-tagged JDP2, recombinant Histone 3 (left), or Histone 4 (right), and Flag antibody-immunoprecipitated lysates from CPT (10 μM, 1 h)-treated cells transfected with 0, 1.0, and 5.0 μg of a Flag-tagged β-catenin plasmid. **g** JDP2 DNA-binding activity analyzed using an EMSA assay were examined in the indicated cells treated with or without CPT (10 μM, 1 h) (upper), or in CPT (10 μM, 1 h)-treated cells transfected with 0, 1.0, and 5.0 μg of a Flag-tagged β-catenin plasmid (middle), or in CPT (10 μM, 1 h)-treated cells transfected with or without β-catenin siRNA (lower). OCT-1 served as the loading control. Each error bar in panels **a** and **b** represents the mean ± SD of three independent experiments. *$P < 0.05$. Student's two-tailed *t* test. Source data of Fig. 4a and b are provided as a Source Data file

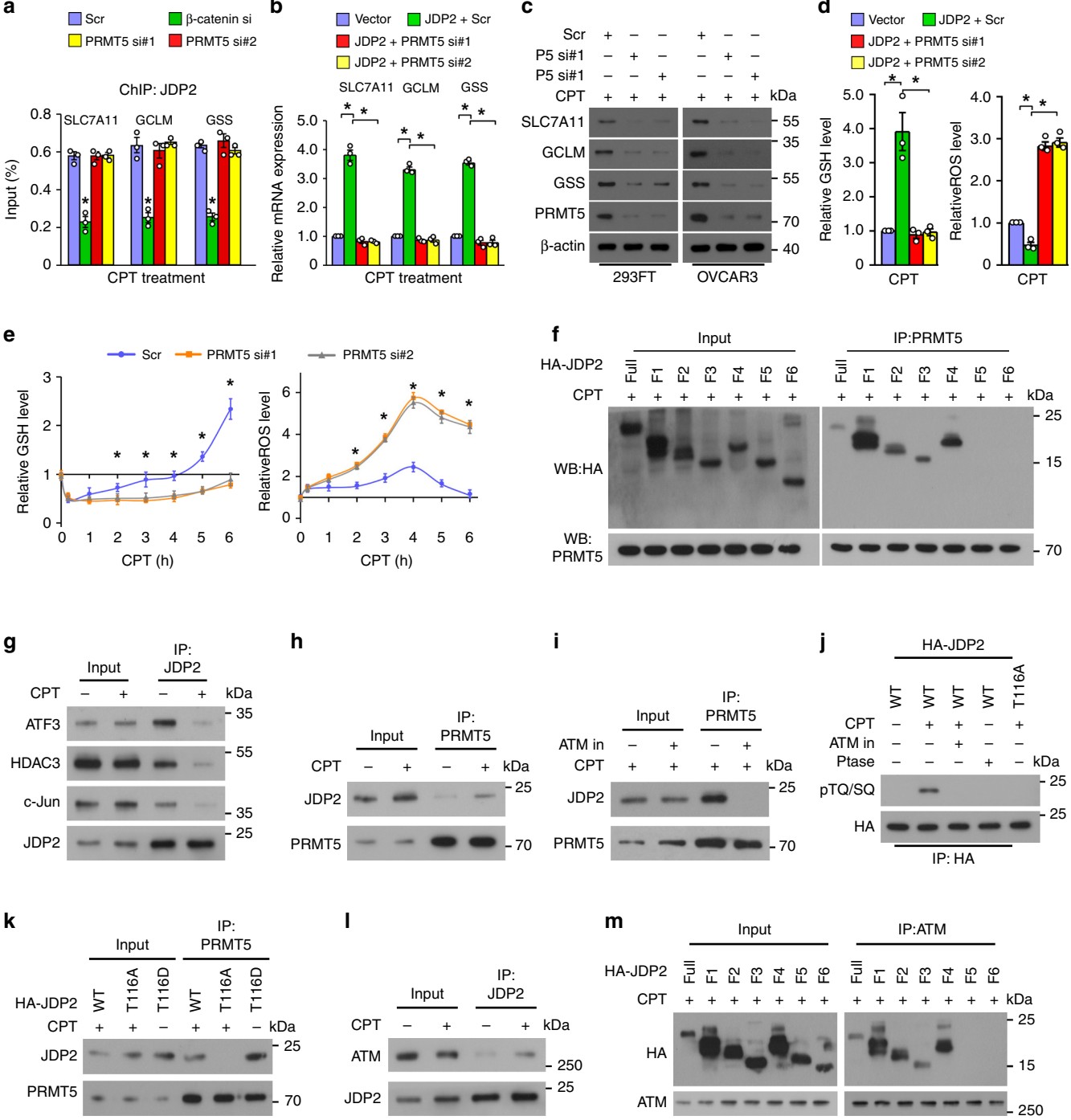

**Fig. 5** PRMT5 promotes β-catenin/JDP2-activated glutathione metabolism. **a**, **b** The ChIP assay and qRT-PCR analysis of the enrichment of JDP2 on the promoters (**a**) and mRNA expression (**b**) of *SLC7A11*, *GCLM*, and *GSS* in CPT (10 μM, 1 h)-treated cells. **c** IB analysis of protein expression of SLC7A11, GCLM, and GSS in the CPT (10 μM, 4 h)-treated cells transfected with scramble or PRMT5 siRNA(s). **d** Relative expression levels of GSH (left) and ROS (right) in the indicated cells. **e** Relative expression of GSH (left) and ROS (right) were examined in the indicated cells. **f** Co-IP assays using anti-PRMT5 antibody were performed in the CPT (10 μM, 1 h)-treated cells transfected with truncated JDP2 fragments (right). **g** IP assays using anti-JDP2 antibody were performed in the indicated cells treated with or without CPT (10 μM, 1 h), and IB analysis of the expression of JDP2, AFT3, c-Jun, and HDAC3. **h** IP assays using anti-PRMT5 antibody were performed in the indicated cells treated with or without CPT (10 μM, 1 h), and IB analysis of expression of JDP2 and PRMT5 was performed. **i** IP assays using anti-PRMT5 antibody were performed in indicated cells treated with CPT (10 μM, 1 h) with or without the ATM inhibitor KU55933 (10 μM, 1 h) pretreatment, and IB analysis of the expression of JDP2 and PRMT5. **j** IP/IB assays analysis of HA-tagged JDP2 and phosphorylation of TQ/SQ in the indicated cells. **k** IP assays using anti-PRMT5 antibody were performed in the indicated cells treated with or without CPT (10 μM, 1 h), and IB analysis of JDP2 and PRMT5. **l** IP assays using anti-JDP2 antibody were performed in the indicated cells treated with or without CPT (10 μM, 1 h), and IB analysis of JDP2 and ATM. **m** co-IP assays were performed using anti-ATM antibody in the CPT (10 μM, 1 h)-treated cells transfected with truncated JDP2 fragments. Each error bar in panels **a**, **b**, **d**, and **e** represents the mean ± SD of three independent experiments. *P < 0.05. Student's two-tailed *t* test. Source data of Fig. 5a, b, d–e are provided as a Source Data file

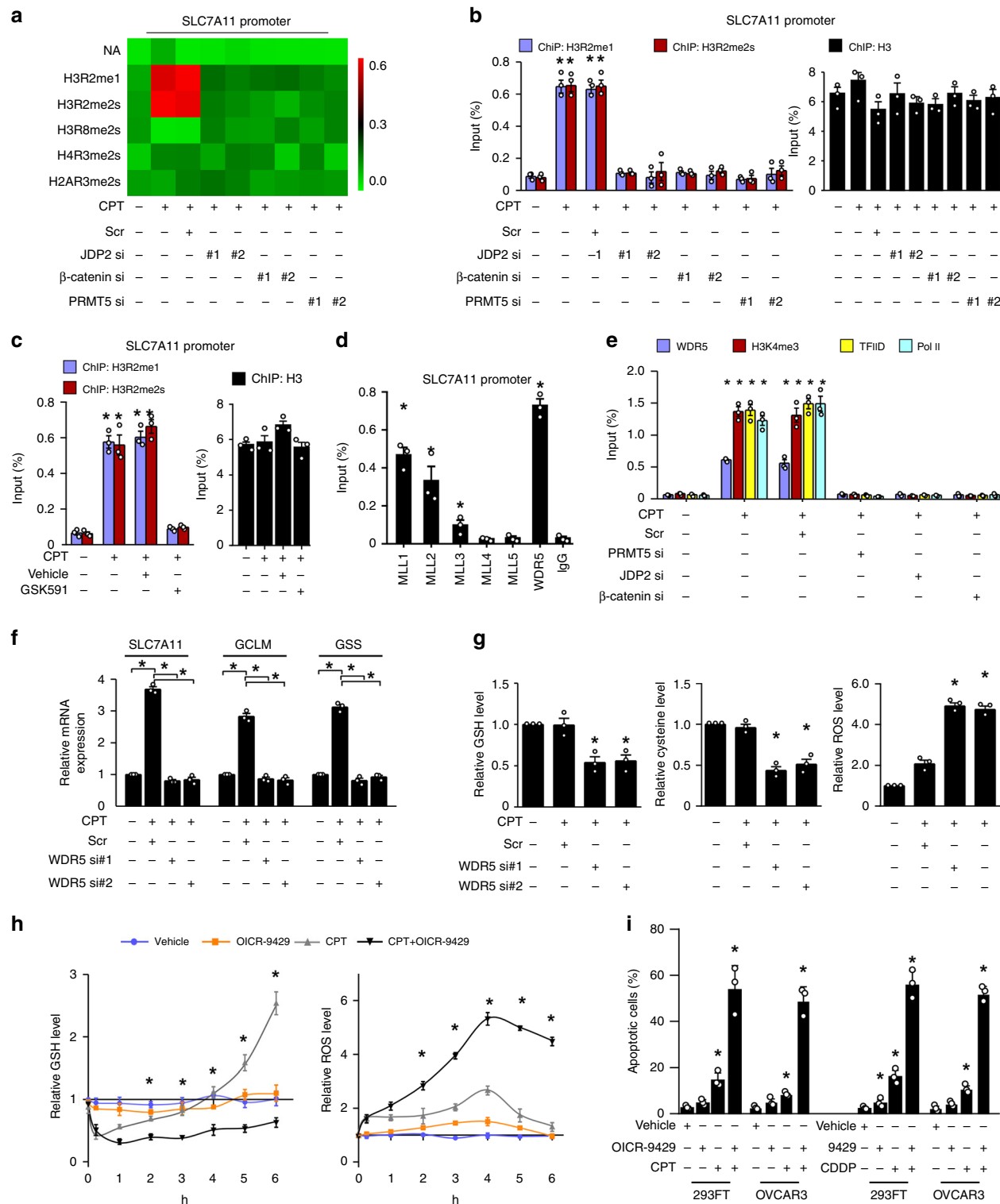

genotoxic treatment by reprogramming GSH metabolism and inhibiting ROS-induced apoptosis.

**OICR-9429 enhances sensitivity of genotoxic drugs**. Our results indicated that antagonism of the WDR5-MLL interaction via small-molecule OICR-9429 led to decreased GSH levels, but increased ROS production (Fig. 6h). Therefore, we further examined the therapeutic effect of combined OICR-9429 and genotoxic chemotherapeutics on cancer using an in vivo patient-

derived xenograft (PDX) model established with two clinical ovarian cancer tissues. As shown in Fig. 10a, b, treatment with OICR-9429 significantly enhanced the sensitivity of ovarian cancer to Topotecan chemotherapy, as indicated by the decreased tumor volume and mass. Prominently, compared with Topotecan alone-treated tumors, combined Topotecan/OICR-9429 treatment resulted in a significant reduction in the GSH concentration and led to a lethal increase in the Topotecan-induced ROS levels in the PDX tumors, which displayed further elevated levels of activated-PARP1 and -Caspase 3 and increased numbers of

**Fig. 6** PRMT5-mediated histone H3R2 methylation contributes to genotoxic stress-induced GSH metabolism. **a** ChIP assay analysis of the enrichment of H3R2me1, H3R2me2s, H3R8me2s, H4R3me2s, and H2AR3me12s on the promoter of *SLC7A11* in the indicated cells treated with CPT (10 μM, 1 h). The heatmap represented by pseudocolors was generated using the ChIP-qPCR values, arrayed from green (no enrichment) to red (maximal enrichment), to demonstrate the histone methylarginine code surrounding the promoter of *SLC7A11*. **b**, **c** ChIP assay analysis of the enrichment of H3R2me1 and H3R2me2s on the promoter of *SLC7A11* in the indicated cells treated with CPT (10 μM, 1 h) (**b**) or PRMT5 inhibitor GSK591 (5 μM, 1 h) (**c**). ChIP-qPCR of Histone 3 served as the control. **d**, **e** ChIP assays analyses of enrichment of MLL1, MLL2, MLL3, MLL4, MLL5, and WDR5 (**d**) or WDR5, H3K4me3, transcriptional factor IID (TFIID), and polymerase II (**e**) on the promoter of *SLC7A11* in CPT (10 μM, 1 h)-treated cells. Anti-IgG antibody served as the control. **f** Relative mRNA expression of *SLC7A11*, *GCLM*, and *GSS* in the scramble- or WDR5 siRNA(s)-transfected cells treated with or without CPT (10 μM, 1 h), as quantified by qRT-PCR analysis. *GAPDH* serve as the loading control. **g** Relative expression levels of GSH (left), cysteine (middle), and ROS (right) were examined in scramble- or WDR5 siRNA(s)-transfected cells treated with or without CPT (10 μM, 4 h). **h** Relative levels of GSH (left) and ROS (right) in vehicle-, or OICR-9429 (a WDR5 inhibitor, 5 μM, 4 h)-, or CPT (10 μM, 4 h), or OICR-9429 (10 μM, 4 h) plus CPT (10 μM, 4 h)-treated cells at the indicated times. **i** Quantification of the apoptotic index in the indicated cells treated with vehicle-, or OICR-9429 (10 μM)-, or CPT (10 μM), or OICR-9429 (10 μM) plus CPT (10 μM), as analyzed by an Annexin-V assay. +: treatment, −: untreatment. Each error bar in panels **a**–**i** represents the mean ± SD of three independent experiments. *$P < 0.05$. Student's two-tailed *t* test. Source data of Fig. 6b–i are provided as a Source Data file

apoptotic cells (Fig. 10c–e). Importantly, we also observed a similar promotive effect of OICR-9429 on the anti-tumor activity of CDDP, another widely used genotoxic chemo drug (Fig. 10a–e). Taken together, these results showed that a combination of classical genotoxic chemotherapeutics with OICR-9429 might represent a strategy in the fight against cancer by improving the therapeutic outcome (Fig. 10f).

## Discussion

Reduced glutathione (GSH), which functions as both a nucleophile and a reductant, plays roles in various cellular processes through regulation of the thiol-redox status. GSH could effectively scavenge ROS (e.g., hydroxyl radical, lipid peroxyl radical, superoxide anion, and hydrogen peroxide) via nonenzymatic reduction or eliminate hydroperoxides required enzymatic catalysis[8–10]. Hence, maintaining or reestablishing intracellular GSH homeostasis is fundamental for cellular physiological functions, such as cell survival and tissue regeneration. Several mechanisms have been reported whereby cells maintain their GSH redox state in response to oxidative stress, such as de novo synthesis and salvage pathways[11–15]. Interestingly, previous studies have documented that the reduced GSH levels in genotoxic stress-treated cells swiftly restored to their original level, and were even further elevated a few hours later[17,18]. However, the mechanism underlying how cells subjected to genotoxic stress reestablish GSH homeostasis remains unclear. In this study, we found that genotoxic stress-activated β-catenin signaling, which played a vital role in rapidly restoring GSH metabolism and promptly eliminating genotoxic stress-induced ROS. Therefore, our results revealed a mechanism for genotoxic stress-induced restoration of redox homeostasis.

The β-catenin signaling pathway plays vital roles in regulating embryonic development, stem cell maintenance, tissue homeostasis, and the progression and development of cancer via TCF-dependent and -independent mechanisms[19–21]. Typically, Wnt-ligand-dependent activation of β-catenin signaling results in the interaction of β-catenin with TCF/LEF factors and activation of TCF-dependent transcription. Interestingly, several studies have documented that $H_2O_2$ treatment-induced ROS-dependent signaling could shift β-catenin binding from TCF to forkhead box O proteins (FOXOs) and induce FOXO-mediated transcription, resulting in removal of $H_2O_2$ via upregulation of manganese superoxide dismutase and catalase and decreased β-catenin/TCF transcriptional activity[22,23], which suggested that β-catenin signaling plays roles in the reduction of ROS via a TCF-independent mechanism. Meanwhile, it was also reported that genotoxic stress-induced poly-ADP-ribosylated PARP1 and upregulated Ku70, which competed for the interaction of TCF4 with β-catenin

and reduced β-catenin/TCF transcriptional activity[21,31]. However, the biological role of β-catenin signaling in response to genotoxic stress remains unclear. In the present study, we observed that, unlike $H_2O_2$ treatment, genotoxic stresses did not increase the β-catenin/FOXOs interaction, suggesting that the effect of β-catenin signaling on the reduction of genotoxic stress-induced ROS might be through other mechanisms. We further demonstrated that genotoxic stresses induced the formation of β-catenin/JDP2/PRMT5 complex, which facilitated the restoration of GSH homeostasis via transcriptional upregulation of multiple genes in the GSH-metabolic cascade, resulting in elimination of genotoxic stress-induced ROS. Therefore, our results revealed a mechanism by which β-catenin signaling maintains redox homeostasis in genotoxic stress-treated cells. However, it is worth to note that expression level of four TCF/LEF factors individually silenced in this study was still detectable. It would be better to knockout all four TCF/LEF factors simultaneously via CRISPR-Cas9 system to further rule out the possibility of functional redundancy of these TCF/LEF factors on GSH regulation.

Human JDP2, originally identified as an AP-1 repressor, is involved in the transcriptional repression of TRE-dependent and CRE-dependent genes via heterodimerization with c-Jun or ATF-2[25,26,32,33]. Further studies demonstrated that JDP2-mediated transcriptional repression acts via distinct mechanisms, including decreasing histone acetylation through associating with histone deacetylases (HDACs), inhibiting p300-mediated acetylation by directly interacting with the core histones, or by promoting supercoiling into circular DNA in the presence of core histones[25,26]. Meanwhile, JDP2 also acts as a transcriptional activator, such as contributing to the promotion of progesterone receptor- or sRANKL (receptor activator of nuclear factor kappa B ligand)-mediated transcriptional activation[34,35]. However, the precise mechanism of JDP2-mediated transcriptional activation remains unknown. In this study, we validated that genotoxic stress-triggered JDP2 formed a complex with β-catenin and PRMT5, which inhibited the binding of JDP2 to histones and HDACs, thereby acting as transcriptional co-activator complex that upregulated multiple GSH-metabolic genes. Therefore, our results unveiled a mechanism by which JDP2-mediated transcriptional activation. Contrastingly, JDP2 was reported to play a role in the antioxidant response through an association with the NF-E2-related factor 2 (NRF2)/ MAF BZIP transcription factor K (MAFK) complex[36]. However, the mechanism by which JDP2-reduced ROS production has not been clarified. In this study, we demonstrated that JDP2 was enriched at gene promoters and transcriptionally upregulated multiple GSH-metabolic genes upon genotoxic stress, resulting in restoration of GSH metabolism and a reduction in genotoxic stress-induced ROS, consequently leading to resistance of cancer to genotoxic

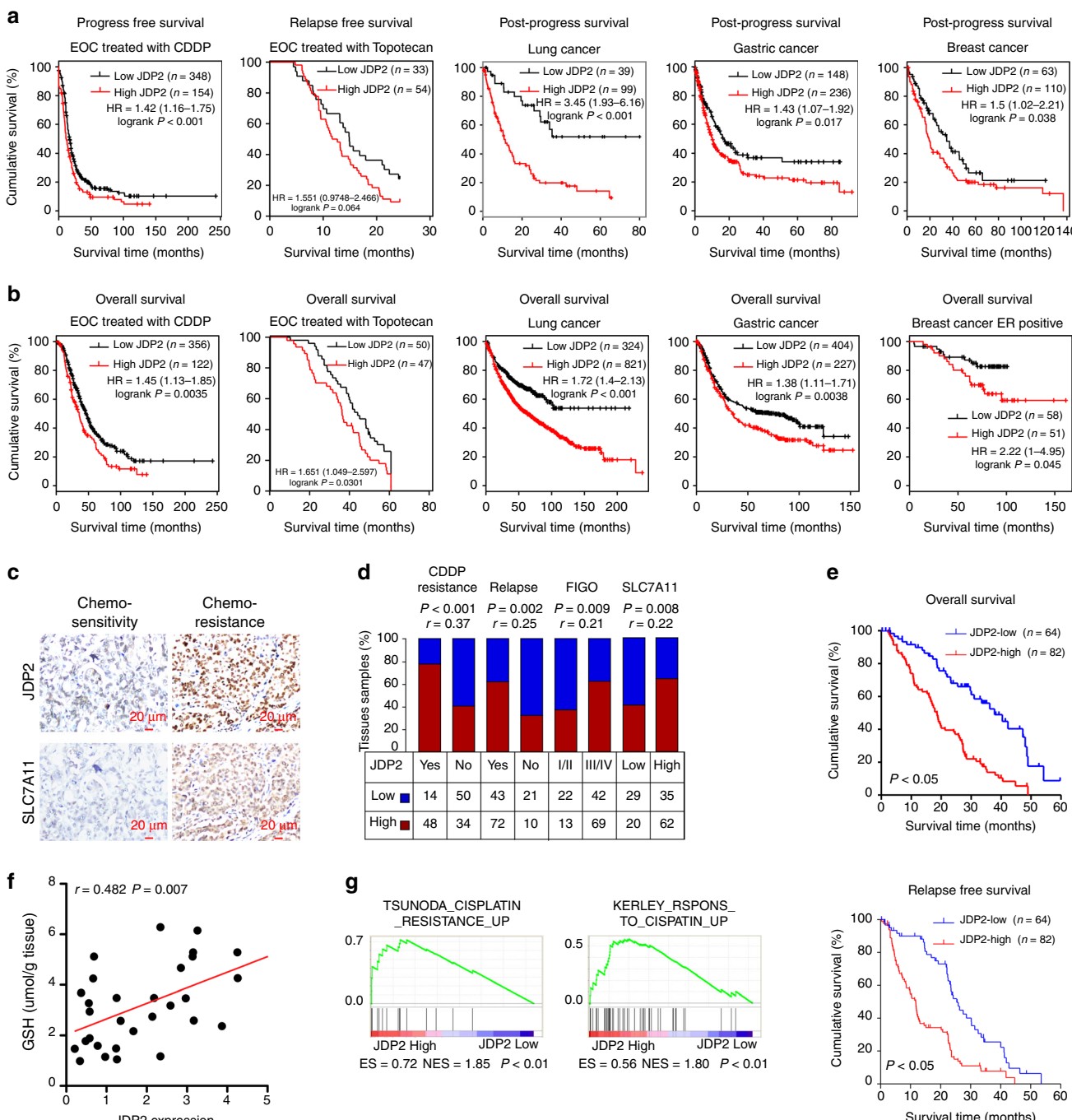

**Fig. 7** JDP2 level correlates with poorer survival of patients with cancer. **a**, **b** Online Kaplan–Meier plotter analysis revealed that patients with ovarian, lung, gastric, or breast cancer exhibiting high JDP2 expression had significantly shorter progression-free survival and shorter overall survival than patients with low JDP2 expression ($P < 0.05$, log-rank test; $n =$ the indicated biologically independent samples). **c** Representative images of JDP2 and SLC7A11 in chemo-sensitive and chemo-resistant ovarian cancer tissues ($n = 146$). Scale bar, 20 μm. **d** Positive correlation of JDP2 levels with CDDP resistance ($P < 0.001$; $r = 0.37$), relapse ($P = 0.002$; $r = 0.25$), FIGO stage ($P = 0.009$; $r = 0.21$), and SLC7A11 expression ($P = 0.008$; $r = 0.22$) in ovarian cancer tissues ($n = 146$). Spearman rank correlation analysis. **e** Kaplan–Meier analysis of 5-year overall survival (upper) and 5-year disease-free survival (lower) for patients with ovarian cancer stratified by low versus high expression of JDP2 (log-rank test; $P < 0.05$, $P < 0.05$, respectively; $n = 146$). Quantification of IHC using the staining index (see Supplementary Materials and Methods). Samples with an SI ≥ 8 were determined as high expression, and samples with an SI < 8 were determined as low expression ($n =$ the indicated biologically independent samples). **f** Positive correlation between JDP2 expression and GSH levels ($P = 0.007$; $r = 0.482$) in 30 freshly collected ovarian cancer tissues. two-tailed Spearman test. **g** Gene set enrichment analysis (GSEA) plot showing that JDP2 expression correlated positively with cisplatin-resistance-activated gene signatures (TSUNODA_CISPLATIN_RESISTANCE_UP) in published gene expression profiles of patients with ovarian cancer (NCBI/GEO/GSE66957, $n = 69$) and in gene expression profiles of patients with breast cancer (TCGA, $n = 1092$)

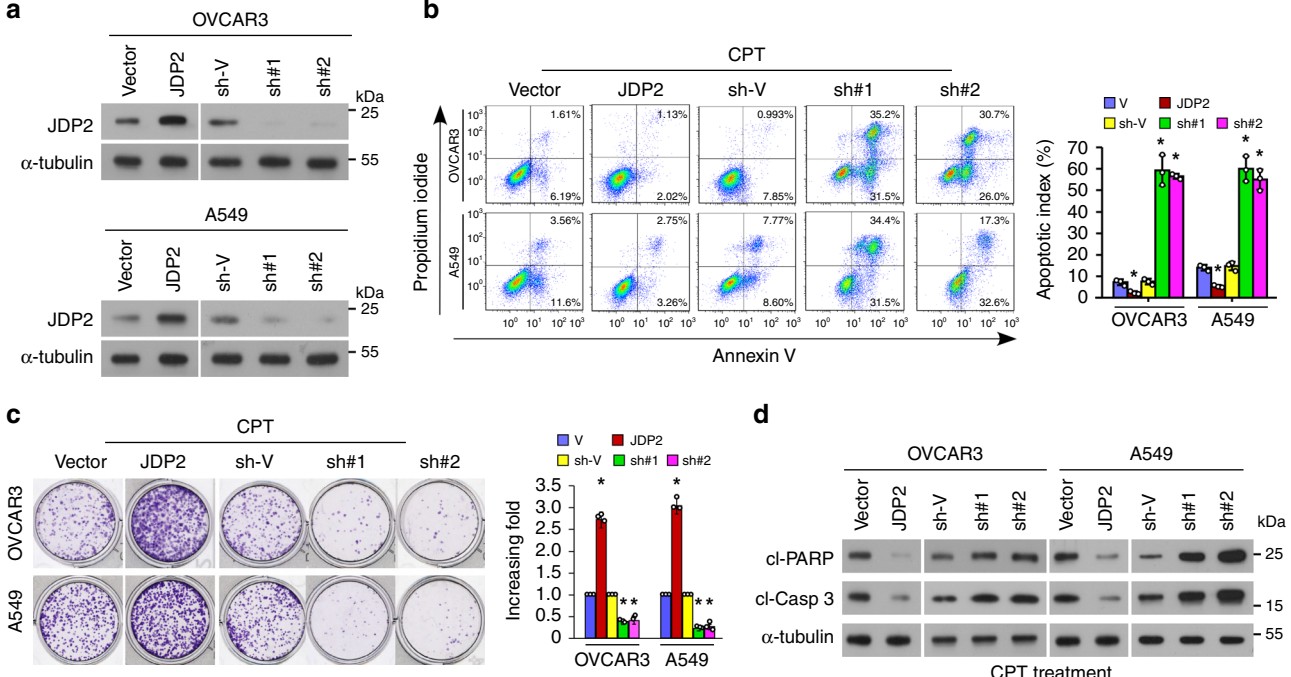

**Fig. 8** JDP2 confers resistance to genotoxic stress on cancer cells in vitro. **a** IB analysis of JDP2 expression in the indicated cells transduced with vector, JDP2, the shRNA-vector, or JDP2 shRNA(s); α-tubulin served as the loading control. **b** FACS analysis of annexin-V staining (left) and quantification (right) of indicated cells treated with Vehicle or CPT (10 μM) after 12 h. **c** Representative images (left) and quantification (right) of the colony number of the indicated cells treated with CPT (10 μM), as determined by a colony-formation assay. **d** IB analyses of expression of cleaved-PARP1 and cleaved-caspase 3 in the CPT (10 μM, 12 h)-treated cells. α-tubulin served as the loading control. Each error bar in panels **b** and **c** represents the mean ± SD of three independent experiments. *P < 0.05. Student's two-tailed t test. Source data of Fig. 8b, c are provided as a Source Data file

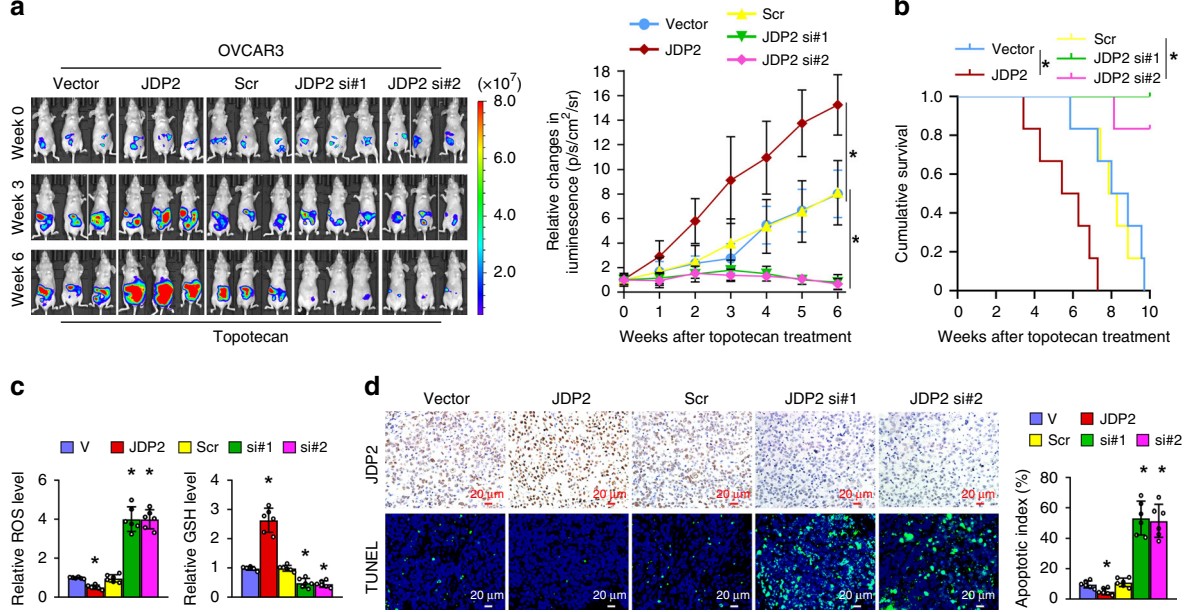

**Fig. 9** JDP2 confers resistance to genotoxic stress on cancer cells in vivo. **a** Representative images of tumor-bearing nude mice inoculated intraperitoneally with the indicated cells in response to Topotecan chemotherapy at the indicated times (left), and the relative change in the bioluminescence signal of intraperitoneal tumors in nude mice in response to Topotecan chemotherapy (right). n = 6 animals per group. **b** Kaplan–Meier survival of mice inoculated intraperitoneally with the indicated cells. n = 6 animals per group. **c** Relative levels of GSH (left) and ROS (right) in the indicated xenograft tumors. **d** IHC staining of JDP2 expression and TUNNEL analysis (left) and quantification (left) of the apoptotic index in the indicated xenograft tumors. Scale bar, 20 μm. Each error bar in panels **c** and **d** represents the mean ± SD of three independent experiments. *P < 0.05. Student's two-tailed t test. Source data of Fig. 9a–d are provided as a Source Data file

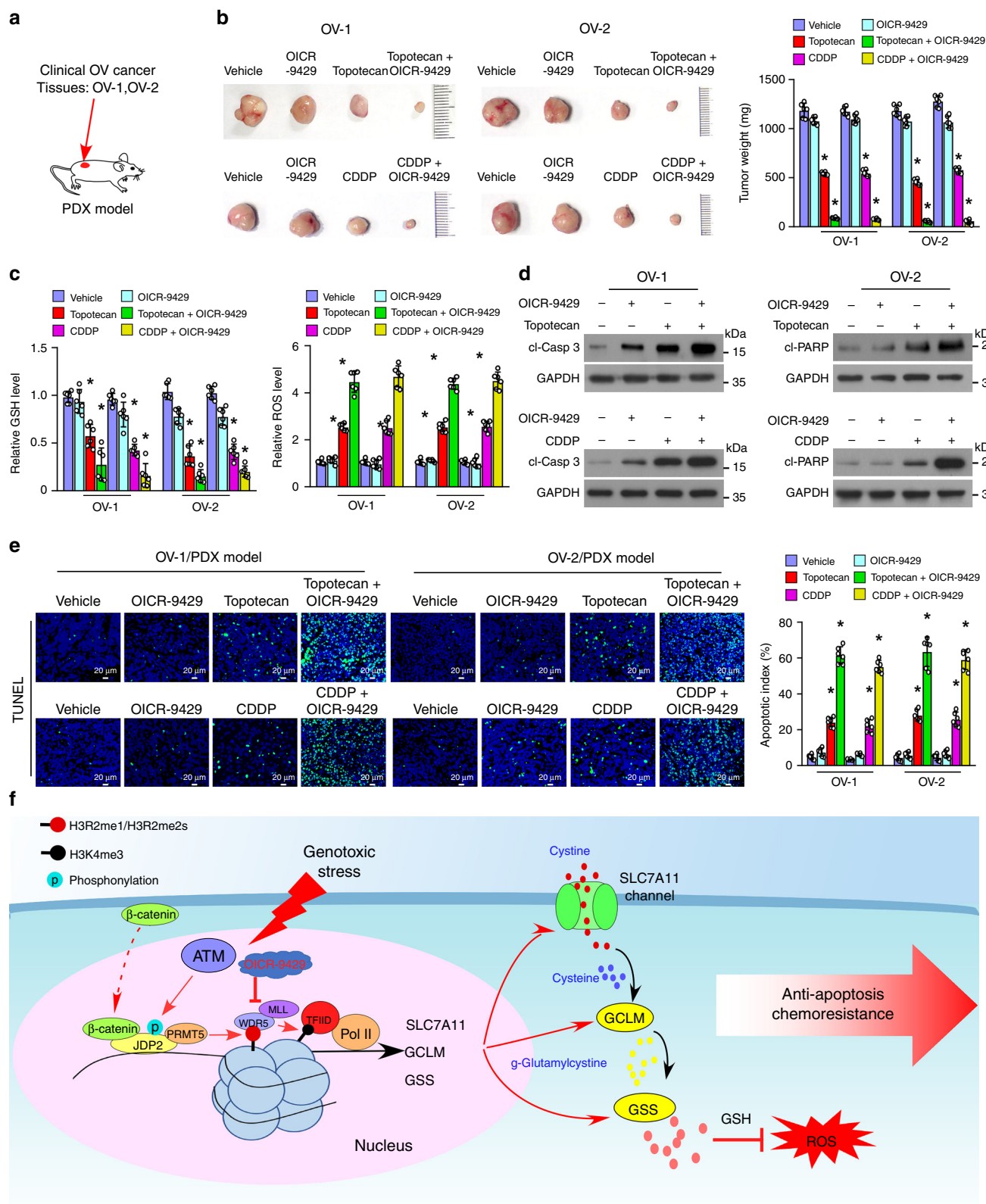

chemotherapeutics. Thus, our results also uncovered a plausible mechanism for the JDP2-mediated maintenance of ROS homeostasis and chemoresistance.

Redox-dependent posttranslational modification (PTM), such as S-glutathionylation of sulfur-containing amino acids, controls a wide range of intracellular protein activities and is involved in the response of oxidative stress[37,38]. For instance, ROS-induced cysteine S-glutathionylation promoted TAZ stability that was critical for ROS-mediated transactivation of TAZ, and oxidative stress-induced S-glutathionylation of mitochondrial thymidine kinase 2 (TK2) have significant impacts on TK2 degradation and mitochondrial DNA precursor synthesis[39,40]. Herein, we

**Fig. 10** WDR5 inhibitor OICR-9429 enhances the sensitivity of ovarian cancer to genotoxic chemotherapeutics in vivo. **a** A PDX model was established by inoculating with two freshly collected clinical primary ovarian cancer tissues, OV-1 and OV-2. **b** Representative pictures (left) and weight (right) of xenograft tumors in response to the indicated chemotherapy. Left, upper: chemotherapy with Vehicle, OICR-9429 (3 mg/kg), Topotecan (10 mg/kg), or OICR-9429 (3 mg/kg) combined with Topotecan (10 mg/kg). Left, lower: chemotherapy with Vehicle, or OICR-9429 (3 mg/kg), CDDP (5 mg/kg), or OICR-9429 (3 mg/kg) combined with CDDP (5 mg/kg). $n = 6$ animals per group. **c** Relative levels of GSH (left) and ROS (right) in the indicated chemotherapy-treated xenograft tumors. $n = 6$ animals per group. **d** IB analysis of the level of cleaved-caspase 3 and cleaved-PARP1 in the indicated chemotherapy-treated xenograft tumors. GAPDH served as the loading control. **e** Representative images (left) and quantification (right) the apoptotic rate the indicated chemotherapy-treated xenograft tumors (left). Scale bar, 20 μm. $n = 6$ animals per group. **h** Hypothetical model illustrating that the genotoxic stress-triggered β-catenin/JDP2/PRMT5 complex plays a vital role in reestablishing glutathione homeostasis and eliminating chemoradiotherapy-induced ROS reduction, resulting in anti-apoptosis and chemoradioresistance, consequently leading to poor clinical prognosis of cancer. Each error bar in panels **b**, **c**, and **e** represents the mean ± SD of three independent experiments. *$P < 0.05$. Student's two-tailed $t$ test. Source data of Fig. 10b, c, and **e** are provided as a Source Data file

demonstrated that β-catenin formed a complex with ATM phosphorylated-JDP2 and PRMT5 that associated with the promoters of multiple genes in the GSH-metabolic cascade, resulted in reestablishing GSH homeostasis upon genotoxic stress. However, we found that genotoxic stress did not induce either S-glutathionylational modification or stabilization of either JDP2, or β-catenin, or ATM, or PRMT5 (Supplementary Fig. 7a, b), which provided further evidence that genotoxic stress-triggered β-catenin/JDP2/PRMT5 complex facilitated reestablishing glutathione homeostasis via de novo GSH synthesis.

Genotoxicity-based chemotherapy and radiotherapy, which induce severe DNA damage, ROS production, and apoptosis of cancer cells, remain the standard first-line treatments for most human cancers. For instance, Sen et al. have previously reported that CPT-induced ROS elevation, especially the level of superoxide radical and hydroxyl radical, promoted apoptotic cell death via dysfunction of cellular respiration and mitochondrial hyperpolarization[41]. Marullo et al. reported that CDDP treatment upregulated ROS levels in cancer cells were dependent on the mitochondria instead of nuclear DNA damage signaling in DU145 and DU145ρ° cells (lacking mitochondrial DNA)[42]. Furthermore, hydroxyl radical scavenger was found to be essential for ameliorating the nephrotoxicity following CDDP chemotherapy via protecting mitochondria and preventing oxidative stress[43]. Consistently, exposing cells to either γ-radiation or α-particles significantly enhanced cellular ROS levels, such as superoxide anions and hydrogen peroxide, via inducing a reversible mitochondrial permeability transition that attributable to normal cell metabolism[44,45]. Therefore, genotoxic stress provoked by inhibitor of DNA topoisomerase, or chemotherapeutic drug (CDDP), or γ-radiation could induce multiple types of ROS. On the other hand, several studies have provided evidence that depletion of GSH could potentiate apoptosis provoked by CPT, or CDDP, or ionizing radiation. These results suggest that cytotoxicity provoked by these reagents were linked to intracellular GSH alteration, which was related with chemoresistance and radioresistance and cancer progression[46–48]. Thus, exploring the mechanism underlying the reprogramming of GSH metabolism in genotoxic stress may aid the identification of therapeutic targets for cancer and avoid the side effects of chemoradiotherapy. According to our finding that genotoxic stress-triggered β-catenin/JDP2/PRMT5 complex formation resulted in WDR5/MLL methyltransferase complex-mediated restoration of GSH homeostasis, we demonstrated that combined treatment with OICR-9429, an antagonist of the WDR5-MLL interaction, could inhibit the GSH-metabolism reestablishing and lead to a lethal increase in the already-elevated levels of ROS in genotoxic chemotherapeutics-treated cancer cells. Therefore, understanding the precise mechanism underlying genotoxic stress-induced reestablishing redox homeostasis would be not only benefit the treatment of a large group of cancer patients, but also would

increase our knowledge of the biological basis of cancer progression.

## Methods

**Cell lines and cell culture**. The OVCAR3 ovarian cancer cell line, A549 lung cancer cell line, MCF7 breast cancer cell line, and human embryonic kidney 293FT cell line were obtained from American Type Culture Collection (ATCC, Manassas, VA, USA) and were grown in the DMEM medium (Gibco, Grand Island, NY, USA) supplemented with 10% fetal bovine serum (Gibco, Grand Island, NY, USA). All the cell lines have been tested for mycoplasma contamination, and were authenticated by short tandem repeat (STR) fingerprinting at Medicine Lab of Forensic Medicine Department of Sun Yat-Sen University (China).

**Plasmids, retroviral infection, and transfection**. The ORFs of JDP2 (Full, T116A mutation and T116D mutation), β-catenin, PRMT5 and truncated-JDP2 and -β-catenin fragments were cloned into pCDNA3 or pSin-EF2-vector. ShRNAs targeting JDP2 were cloned into the pSuper Retro viral vector. The TCF/LEF1 reporter plasmids contain wild-type (CCTTTGATC; TOP flash) or mutated (CCTTTGGCC; FOP flash). Transfection of siRNAs or plasmids was performed using the Lipofectamine 3000 reagent (Thermo Fisher Scientific, Waltham, MA, USA) according to the manufacturer's instruction. All siRNA oligonucleotides are listed as Supplementary Data 3. Stable cell lines-expressing JDP2 and JDP2 shRNA (s) were generated via retroviral infection using 293FT cells, and stable cell lines-expressing JDP2 or JDP2 RNAis were selected for 10 days with 0.5 μg/ml puromycin 48 h after infection.

**RNA extraction, reverse transcription, and real-time PCR**. The total RNA was extracted from indicated cell using the Trizol (Life Technologies) reagent according to the manufacturer's instruction. Real-time reverse transcription–polymerase chain reaction (PCR) primers and probes were designed with the assistance of the Primer Express v 2.0 software (Applied BioSystems, Foster, CA, USA). Expression data were normalized to the geometric mean of housekeeping gene *GAPDH* to control the variability in expression levels and calculated as $2^{-[(C_t \text{ of gene}) - (C_t \text{ of GAPDH})]}$, where $C_t$ represents the threshold cycle for each transcript. All primers are listed as Supplementary Data 3.

**Chemical reagents**. Camptothecin (CPT), Cisplatin (CDDP), Topotecan, ATM inhibitor (KU55933), and WDR5 inhibitor (OICR-9429) PRMT5 inhibitor (GSK591) were purchased from MedChemexpress (Monmouth Junction, NJ, USA). Human recombinant His-JDP2 protein was purchased from Abcam (Cambridge, MA, USA).

**Immunoblotting analysis (IB)**. IB was performed according to a standard protocol with the following antibody: anti-β-catenin (610154, 1:500), anti-α-catenin (610194, 1:500), and anti-PARP1 (611039,1:500) antibodies were purchased from BD company (franklin lakes, NJ, USA). Anti-JDP2 (ab40916,1:500), anti-PRMT5 (ab109451,1:1000), anti-SLC7A11 (ab37185,1:200), anti-GCLM (ab124827, 1:500), anti-TCF4 (ab76151, 1:1000), anti-p-SQ/TQ ATM/ATP (ab130947, 1:500), anti-H3 (ab1791, 1:1000), anti-H4 (ab10158, 1:500), anti-ATM (ab81292,1:500), anti-ATF3 (ab207434, 1:500), anti-HDAC3(ab32369, 1:1000), anti-FOXO3 (ab12162,1:1000), anti-TCF1 (ab30961, 1:1000), anti-LEF1 (ab217378, 1:500), anti-cleaved PARP1 (ab32064, 1:500) and anti-GSH (ab19534, 1:500) antibodies were purchased from Abcam (Cambridge, MA, USA). Anti-c-Jun (#9165,1:500), anti-cleaved Caspase 3 (#9661 S, 1:500), anti-His (#12698 s, 1:500), anti-TCF3 (#28831, 1:500), anti-SMARCA4 (#52251, 1:500), and anti-HNRNPA2B1 (#9304, 1:500) antibodies were purchased from cell signaling technology (Danvers, MA, USA). Anti-WDR5 antibody (07-706, 1:500) was purchased from Millipore (Billerica, MA, USA). Anti-Flag (F3165, 1:1000) and anti-HA (H9658, 1:1000) antibodies were purchased from Sigma-Aldrich (St. Louis, MO, USA). Membranes were stripped and re-probed

with an anti-α-tubulin antibody (ab7291, 1:3000), or anti-GAPDH antibody (ab181602, 1:3000), or anti-β-actin antibody (ab28925, 1:3000) as a protein loading control. Uncropped images of Immunoblotting were provided in the Supplementary Fig. 8 in the Supplementary Information.

**Immunofluorescence (IF) staining**. IF staining was carried out on tumor cell chamber slide cultures (Thermo Fisher Scientific), and followed by the antibody: β-catenin (BD, 610154, 1:200) and anti-JDP2 (ab40916, 1:500) antibodies. The secondary antibody was goat anti-rabbit IgG (H + L) conjugated with Alexa Fluor 568 (Thermo Fisher Scientific). Then cells were mounted with Antifade Mountant with DAPI (Thermo Fisher Scientific). The images were captured using the AxioVision Rel.4.6 computerized image analysis system (Carl Zeiss, Jena, Germany).

**Immunohistochemistry (IHC)**. IHC analysis was performed to determine altered protein expression in 146 paraffin-embedded ovarian cancer tissues with Rabbit anti-JDP2 (ab40916, 1:500) and anti-SLC7A11 (ab37185, 1:200) antibodies overnight at 4 °C. The degree of immunostaining of formalin-fixed, paraffin-embedded sections were reviewed and scored separately by two independent pathologists uninformed of the histopathological features and patient data of the samples. The scores were determined by combining the proportion of positively stained tumor cells and the intensity of staining. The scores given by the two independent pathologists were combined into a mean score for further comparative evaluation. Tumor cell proportions were scored as follows: 0, no positive tumor cells; 1, < 10% positive tumor cells; 2, 10–35% positive tumor cells; 3, 35–75% positive tumor cells; 4, > 75% positive tumor cells. Staining intensity was graded according to the following standard: 1, no staining; 2, weak staining (light yellow); 3, moderate staining (yellow brown); 4, strong staining (brown). The staining index (SI) was calculated as the product of the staining intensity score and the proportion of positive tumor cells. Using this method of assessment, we evaluated protein expression in benign esophageal epithelia and malignant lesions by determining the SI, with possible scores of 0, 2, 3, 4, 6, 8, 9, 12, and 16. Samples with a SI ≥ 8 were determined as high expression and samples with a SI < 8 were determined as low expression. Cutoff values were determined on the basis of a measure of heterogeneity using the log-rank test with respect to overall survival.

**Chromatin fraction**. Briefly, cells were harvested and resuspended in 200 μl of buffer A (10 mM HEPES, [pH 7.9], 10 mM KCl, 1.5 mM MgCl$_2$, 0.34 M sucrose, 10% glycerol, 1 mM dithiothreitol, 0.1% Triton X-100, and protease inhibitor mixture), and incubated for 5 min on ice. The cell pellet was collected by low-speed centrifugation (4 min, 1300 × g, 4 °C). After washed once in buffer A, the cell pellet was lysed in 200 μl of buffer B (3 mM EDTA, 0.2 mM EGTA, 1 mM dithiothreitol, and protease inhibitor mixture) for 10 min on ice. Insoluble chromatin was collected by centrifugation (4 min, 1700 × g, 4 °C), washed once in buffer B, and resuspended in SDS sample buffer.

**Immunoprecipitation and mass spectrometry (MS) analysis**. The chromatin fraction, prepared from CPT (10 μM, 1 h)-treated β-catenin-transduced 293FT cells (5 × 10$^8$), was treated with DNAase (stemcell Technologies) for 1.0 h and then incubated with anti-β-catenin antibody (BD 610154, franklin lakes, NJ, USA) overnight at 4 °C. After then the supernatants were incubated with 20 μl of protein G-agarose beads overnight at 4 °C. The agarose beads were then washed six times with wash buffer (25 mM HEPES [pH 7.4], 150 mM NaCl, 0.5% NP-40, 1 mM EDTA, 2% glycerol, 1 mM PMSF). The eluates were pooled and concentrated in a 10-kDa MW cutoff filter unit (Millipore, Billerica, MA) to a volume of 30 μl. The eluates were denaturation by addition of 10 μl 4 × sample buffer and were subjected to mass spectrometry (MS) analysis.

**Far-western analysis**. Far immunoblotting were performed by using the proteins immunoprecipitated by anti-β-catenin antibody and Human recombinant His-JDP2 protein. Briefly, the proteins were separated by SDS-PAGE, and were transferred onto a PVDF membrane. Membranes were then preincubated in 10% skimmed milk for 1 h at 4 °C. As indicated, recombinant His-JDP2 protein was added at 5 μg/ml and incubated at 4 °C for 18 h. After extensive washing six times with TBST, the membrane was subjected to immunoblotting analysis by indicated antibody.

**Xenografted tumor models**. All of the animal procedures were complied with all relevant ethical regulations for animal testing and research, and the ethical approval was approved by the Sun Yat-sen University Animal Care Committee. Two models were examined in this study. In the subcutaneous patient-derived xenografts (PDX) tumor model, a fragment of freshly tumor isolated clinical ovarian cancer patient tissues were placed subcutaneously (1–3 mm$^3$) underneath the skin of female NOD-SCID IL-2rγ$^{-/-}$ (NSG) mice (4–8 weeks old). Tumor growth was monitored by measurements of tumor diameters once weekly, and the tumor volume was calculated as (larger diameter × smaller diameter$^2$)/2. Recipient mice bearing ~0.2 cm$^3$ size of tumors were intraperitoneally treated with vehicle (control), CDDP (5 mg/kg), Topotecan (10 mg/kg), or OICR-9429 (3 mg/kg) alone, or combined OICR-9429 (3 mg/kg) and Topotecan (10 mg/kg), or combined

OICR-9429 (3 mg/kg) and CDDP (5 mg/kg body weight), three times per week for up to 6 weeks. In the intraperitoneal tumor model, OVCAR3 luciferase expressing cells (5 × 10$^6$) were injected intraperitoneally into female nu/nu nude mice. Recipient mice-bearing tumors when luminescence signal reached > 2 × 10$^7$ p/sec/cm$^2$/sr were treatment with Topotecan (10 mg/kg) for up to 6 weeks. Tumor burden was measured weekly in mice injected with d-luciferin using the IVIS Spectrum In Vivo Imager.

The mice were killed once at the end of treatment, and tumors were harvested and weighed, and prepared for analysis. Tumor sections were used by H&E stained with Mayer's hematoxylin solution, or IHC stained using anti-JDP2 antibody (ab40916,1:500), or stained with TUNEL (In Situ Cell Death Detection Kit, TMR red, Roche Applied Science) and DAPI (Thermo Fisher Scientific) according to manufacturer's protocols. The images were captured using the AxioVision Rel.4.6 computerized image analysis system (Carl Zeiss, Jena, Germany).

**Chromatin immunoprecipitation (ChIP) assay**. The entire procedure was performed with the chromatin immunoprecipitation (ChIPs) assay kit (Upstate/Millipore, Billerica, MA) according to the manufacturer's instructions. Briefly, indicated cells were grown to 70–80% confluence on 100-mm culture dish, and were fixed with 1% formaldehyde to cross-link proteins to DNA. The cell lysates were sonicated to shear DNA into small uniform fragments. Equal aliquots of chromatin supernatants were then immunoprecipitated overnight at 4 °C using anti-JDP2 (ab40916), or anti-β-catenin (BD 610154), anti-PRMT5 (NBP2-19935), anti-TCF4 (ab76151), anti-WDR5 (07-706), anti-H3 (ab1791), anti-H3R2me1 (ab15584), anti-H3R2me2s (ab194684), anti-H3R8me2s (ab130740), anti-H2AR3me2s (ab22397), anti-H4R3me2s (ab5823), anti-H3K4me3 (ab12209), anti-RNAP (ab76123), anti-TAF3(Millipore, 07-1802) antibodies, or an anti-Anti-IgG antibody (a negative control, Millipore, Billerica, MA) respectively, with protein G magnetic beads. Two micrograms of each antibody was used for ChIP per 10$^7$ cells. The cross-linked protein/DNA complexes were collected by magnetic pull down, and then were eluted from beads by elution buffer. After reverse cross-link of protein/DNA complexes to free DNA, PCR was performed using specific primers. All ChIP primers are listed as Supplementary Data 3.

**Preparation of nuclear extracts**. Nuclear fraction was performed by using NE-PER™ Nuclear and Cytoplasmic Extraction Reagents (Thermo Fisher) according to the manufacturer's instructions. The indicated cells were treated with 0.25% trypsin-EDTA, and subsequently harvested at centrifuge at 500 × g for 5 min. The cell pellet was washed with PBS (pH 7.4) 2–3 times and then transferred to a 1.5-mL microcentrifuge tube and centrifuged at 500 × g for 2–3 min. The cell pellet was resuspended in 200 μl ice-cold CER I and vortexed the tube vigorously and incubated on ice for 10 min. The cells were added 11 μl ice-cold CER II to the tube, vortexed for 5 s and centrifuged for 5 min at 16,000 × g. The supernatant (cytoplasmic extract) was immediately transfer to a clean pre-chilled tube. The pelleted nuclei were resuspended in 100 μl ice-cold NER, which contains nuclei and vortexed for seconds every 10 min, for a total of 40 min. The supernatant (nuclear extract) fraction was then centrifuged at ~16,000 × g for 10 min, and then immediately transferred the to a clean pre-chilled tube. Store extracts at −80 °C until use.

**Electrophoretic mobility shift assay (EMSA)**. Electrophoretic mobility shift assay was performed by using the LightShift Chemiluminescent EMSA kit (Thermo Fisher) according to the manufacturer's instruction. To evaluate the effects of genotoxic stress on JDP2 binding to DNA in the nucleus, nuclear extracts from indicated cells were used in an EMSA. The biotin-labeled oligonucleotide probe was synthesized (Thermo Fisher), and unlabeled oligonucleotide with an identical was synthesized as a control. OCT-1 probe was used as a loading control. The sequences of probes are listed in Supplementary Data 3.

**GSH content and cysteine activity assay**. The GSH-Glo™ Glutathione Assay Kit (Promega, Madison, WI, USA) and the cysteine Detection Assay Kit (Sigma-Aldrich, St. Louis, MO, USA) were used to determine the reduced GSH content and cysteine in indicated cells, and the luciferase signal was determined by Spectra Mas M5 (Molecular Devices). Experiments were performed with biological replicates.

**Analysis of cellular ROS**. Intracellular ROS levels were examined using Fluorometric Intracellular ROS Kit (Sigma-Aldrich, St. Louis, MO, USA). Briefly, 1 × 10$^3$ cells were cultured in a 96-well plate, and then cell permeable oxidative fluorescent dye 2',7' dichlorodihydrofluorescein diacetate (H2DCFDA) was added to wells and incubated for 1 h at 37 °C. ROS levels were quantified by measuring fluorescence intensity at excitation and emission wavelength of 490 and 525 nm, respectively using Spectro fluorimeter.

**Detection of 8OHdG**. DNA damage was measured by immunofluorescence with an anti-8-hydroxydeoxyguanosine (8OHdG) monoclonal antibody (Abcam), following the manufacturer's instructions. The secondary antibody was goat anti-mouse IgG2α conjugated with Alexa Fluor 488. Then cells were mounted with Vectashield Mounting Medium with DAPI (Thermo Fisher Scientific). The images

were captured using the AxioVision Rel.4.6 computerized image analysis system (Carl Zeiss, Jena, Germany).

**ChIP-seq analysis**. ChIP-seq assay was performed using Kit according to the manufacturer's instructions. Briefly, chromatin was isolated from β-catenin-transduced 293FT cells treated with CPT (10 μM, 1.0 h). DNA library was labeled for high-throughput sequencing using the Illumina ChIP-seq Library kit following the manufacturer's protocol. The reads were first mapped the human genome sequence (hg19) by using bowtie2 (version 2.2.9). The parameters were set as default, SAMtools (version 0.1.19) were then used to convert files to bam format, sort. Peaks were called using MACS (version 2.1.1)[49–51], with FDR ≤ 0.05. A peak was assigned to the transcriptional start site (TSS) of a RefSeq gene when falling into the surrounding 4 kb ( ± 2 kb). Promoters were defined as 6 kb regions ( ± 3 kb) surrounding the TSS. The promoter peaks analysis, KEGG Pathway, and GO analysis were implemented using KOBAS (version 3.0). $P$-value was calculated by Student's $t$ test.

**RNA-seq analysis**. β-catenin-silenced or control 293FT cells were treated with CPT (10 μM, 4 h), and the total RNA was extracted and purified using the Trizol (Life Technologies) reagent according to the manufacturer's instruction. RNA quantitation and quality control were performed by Bioanalyzer 2100 (Agilent Technologies). Construction of stranded RNA-seq libraries for high-throughput sequencing were done on Illumina HiSeq X Ten following the manufacturer's protocol. RNA-seq reads were mapped to reference genome of Illumina Ensembl genome GRCh37 using HISAT2 version 2.2.9. Mapped reads were summarized for each gene using htseq-count version 0.11.2. Differential expression analysis was implemented using DEGseq version 1.36.1. $P$-value was calculated by Student's $t$ test.

**Patient information and tissue specimens**. This study, which complied with all relevant ethical regulations for work with human participants, was conducted on a total of 146 paraffin-embedded ovarian cancer samples and 30 freshly collected ovarian cancer tissues. The clinical information regarding the samples is summarized in Supplementary Tables 2–3. All patients were received standardized platinum-based chemotherapy. Platinum resistance and sensitive refer to the time-to-relapse within 6 months or after 6 months following completion of platinum-based chemotherapy. The study protocols were approved by the Institutional Research Ethics Committee of Sun Yat-sen University Cancer Center for the use of these clinical materials for research purposes. All Patients' samples were obtained according to the Declaration of Helsinki and each patient signed a written informed consent for all the procedures.

**Luciferase assay**. Ten thousand cells were seeded in triplicate in 48-well plates and allowed to settle for 24 h. One hundred nanograms of luciferase reporter plasmids or the control-luciferase plasmid, plus 5 ng of pRL-TK renilla plasmid (Promega, Madison, WI, USA), were transfected into indicated cells using the Lipofectamine 3000 reagent (Invitrogen, Carlsbad, CA, USA) according to the manufacturer's recommendation. Luciferase and renilla signals were measured 48 h after transfection using the Dual Luciferase Reporter Assay Kit (Promega, Madison, WI, USA) according to a protocol provided by the manufacturer. Three independent experiments were performed, and the data are presented as mean ± SD.

**Cell clonogenic survival assay**. The indicated cells ($2 \times 10^3$) were seeded in six-well culture plates and cultured with fresh median with CPT (10 μM). After incubation for an additional 10 days, the colonies were stained with 1% crystal violet for 30 s after fixation with 10% formaldehyde for 5 min. Colonies of at least 50 cells were quantified by Analysis software (Olympus Biosystems).

**Annexin-V assay**. Apoptotic cells were quantified via the ApopNexin TM FITC Apoptosis Detection Kit (Millipore, Lake Placid, NY), according to the manufacturer's instruction. Indicated treated cells were washed with PBS and then with the Annexin-V binding solution. Subsequently, the cells were added 150 μl of an Annexin-V antibody in binding buffer and incubated for 15 min, and followed by addition of 1.5 μl of PI at 1 mg/ml for a further incubation 5 min. The apoptotic cells were analyzed on a flow cytometer (FACSCalibur; BD Biosciences).

**Statistics**. Statistical tests for data analysis included Fisher's exact test, log-rank test, Chi-square test and Student's two-tailed $t$ test. Bivariate correlations between study variables were calculated by Spearman's rank correlation coefficients. Survival curves were plotted by the Kaplan–Meier method and compared by the log-rank test. The significance of various variables for survival was analyzed by univariate and multivariate Cox regression analyses. Statistical analyses were performed using the SPSS 11.0 statistical software package. Data represent mean ± SD. $P$-values of 0.05 or less were considered statistically significant.

**Reporting summary**. Further information on research design is available in the Nature Research Reporting Summary linked to this article.

## Data availability

All ChIP-seq and RNA-seq data have been deposited in the National Center for Biotechnology Information Sequence Read Archive (SRA) database with BioProject accession code PRJNA543097 (SRA study: SRR9060458) and PRJNA543096 (SRA study: SRR9060518). All data sets reported in this paper are available at the Gene Expression Omnibus with accession number GEO: GSE66957 and The Cancer Genome Atlas, TCGA. The Source data underlying Figs. 1–10, and Supplementary Figs. 1–7 are provided as Source Data file. Uncropped images of Immunoblotting were provided as Supplementary Fig. 8. A reporting summary for this article is available as a Supplementary Information file.

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

## Acknowledgements

This work was supported by Natural Science Foundation of China (No. 81830082, 91740119, 91529301, 81621004, 91740118, 81773106, and 81530082); Guangzhou Science and Technology Plan Projects (201803010098); Natural Science Foundation of Guangdong Province (2018B030311009, 2016A030308002, and 2018B030311060); The Fundamental Research Funds for the Central Universities [No.17ykjc02]; China Postdoctoral Science Foundation (2019M653220).

## Author contributions

L.C., and G.W. designed the experiments and analyzed the data. Z.T., M.T. and D.S. performed the xenograft tumor experiments. G.W., J.Z., M.T., and S.Z. performed in vitro cell line studies. M.T., Y.H. and J.W. performed immunohistochemical and pathological analysis. Z.L., Y.H., S.Z. S.M. and R.Y. performed the ChIP, immunoprecipitation, western blot, and real-time PCR. L.C. and X.W. analyzed ChIP-seq and RNA-seq data. E.S., M.L. and L.S. revised the paper. J.L. supervised the whole study and wrote the paper.

## Additional information

**Competing interests:** The authors declare no competing interests.

