## [Peer Review File · Nature Communications]

Reviewers' comments:

Reviewer #1 (Remarks to the Author):

This is a densely written paper with focus on factors that contribute to regulation of gene clusters relevant to certain component parts of redox homeostasis. In general terms, the English narrative is patchily presented and should be reconsidered by someone with English as a first language. Redox homeostasis is discussed in generic and sometimes vague terms and the term ROS (reactive oxygen species) is used throughout as if it is one entity. The experiments appear to be reliably presented and their interpretation appears reasonable throughout. The biggest concern is the fact that there are few mechanistic considerations as to how ROS mediated chemical reactions are transmitted through the transcriptional regulatory pathways studies to produce the end results reported. For example, the authors do not distinguish hydrogen peroxide, hydroxyl radical, superoxide (or some other species?) or how the oxidative stress stimulates downstream events. As presented, the interference with protein:protein interactions and subsequent effects on protein complexes presumably occurs through ROS mediated changes in protein structures. Is this post-translational modifications (in this case maybe S-glutathionylation?), protein stability or something different? Camptothecin, cisplatin and radiation will produce quite distinct reactive chemical species and these will be dose dependent. Also, cytotoxicities of these agent may not all be directly linked to GSH.

Minor issues.

While the primary focus of this paper is de novo GSH synthesis, GSH homeostasis can also be influenced by salvage pathways, for example through γ -GGT. There is little indication that this is considered.

It is interesting that SLC7A11 is expressed at such high levels in the tumor cell lines used. Normally, significant expression is associated with brain and neurons.

Lines 66/67 do not make sense... "Interestingly, the genotoxic stress-mediated reduction of GSH level could speedily recovery back and even further elevated after a few hours."

Line 279....it is not clear what is meant by "biological free radicals." This also raises the issue that there is little discussion of which type of electrophilic species or which nucleophilic (amino acid?) targets are critical to the observed transcriptional events.

Line 286... "cellular GSH levels would be speedily exhausted and the cells become unable to counteract the ROS-mediated insults, which would lead to irreversible cell degeneration and death." This is a kinetic analysis without data.

Line 311.... "Genotoxic stress, such as DNA-damaging agents, topoisomerase inhibitors, and cisplatin, could sharply induce DNA damage, higher ROS levels, and stronger cytotoxicity compared with that produced by H₂O₂ treatment." Again, this does not really make sense, but presumably the implication is that DNA damage is more relevant to cytotoxicity than hydrogen peroxide. How is this linked to drug induced changes in GSH homeostasis?

Line 343. "However, emerging evidence demonstrated that GSH confers protection against chemoradiotherapy in cancer by reducing genotoxic stress-induced ROS (refs 9, 40, 41)." This statement oversimplifies the depth of existing literature for the field of GSH pathways and drug resistance, and presents two papers on AsO₃ and one that does not seem relevant at all. In classifying ROS as a composite of all chemical species, there is little to distinguish between radiation and all chemotherapeutic agents.

Reviewer #2 (Remarks to the Author):

Cao et al have submitted an interesting paper defining a b-catenin/JDP2/PRMT5 axis in mediating the cellular response to genotoxic stress. The paper is wonderfully detailed in its analysis of mechanism, and the mechanistic insights are interesting and novel. The authors provide correlation data linking JDP2 and poor patient outcome for a number of human cancer parameters. Finally, they demonstrate functional outcomes of manipulating JDP2 in multiple tumor models to demonstrate that targeting JDP2 increases the chemosensitivity of genotoxic chemotherapeutic agents and therefore supports the idea that this approach could be considered as an adjuvant therapy for current chemotherapeutic regimens. I believe the authors have supported their conclusions thoroughly, and will not ask for additional experiments.

My primary concern is that technical standards for data presentation and analysis are not met.

The authors do not include relevant information on the ChIP-seq analysis they performed. Only by reading the Reporting summary can one learn that the ChIP-seq data are from a single replicate with 23 million reads. That might be an adequate number of reads, but it is unclear if that is total reads or mapped reads, or something else. I would think that at minimum there should be a duplicate sample and there should be regression analysis showing the extent of correlation between the datasets. Was input sequenced? Fig legend 1F says peak density was analyzed, but the Reporting Summary makes reference to MACS for peak calling. I find no reference to data obtained via MACS in the manuscript. Inputting the RNA-seq and the ChIP-seq accession numbers for the GEO datasets generated a "not found" message.

The ChIP-seq data set was analyzed for Go terms and for location relative to genes. A very small percentage of b-catenin binding sites were in "promoters", and the authors presented binding relative to TSS sites of genes, but the authors do not indicate how many peaks were identified across the genome. Was it 100 or 10000? Did those peaks correlate with any known cis-element sequences? Are there other data sets for these cells in the literature, perhaps indicating areas of histone modification, or anything else, that could be examined for overlap?

The authors performed MS following IP with b-catenin antibody to identify interacting proteins. They should mention in the text of the Results that this was done using cells transfected with a b-catenin expression vector so that it is immediately understood that the experiment was performed under overexpression conditions. A gel with some of the identified bands is presented along with a representative MS plot to show the sequences identified as JDP2 and PRMT5. There is no discussion of the MS analysis - just a statement that it was performed. No experimental details are provided whatsoever. These should be presented in detail so that those who are interested understand the settings and cutoffs that were used. It is unlikely that the results were limited to the proteins identified in the gel. Shouldn't there be a comprehensive list of the proteins identified, and preferably, the peptide sequences identified by MS, the number of hits per protein, the coverage achieved? That is a pretty basic requirement for IP-MS studies.

In Figure 2b and 2c, and the accompanying supplemental figures, the authors show that siRNA mediated knockdown against all of the candidate interacting proteins do or do not affect expression, factor binding, etc. In the case where effects are seen, they follow-up and present the westerns showing the effectiveness of the siRNA. However, the negative conclusions stating that SMARCA4, PARP, TCF4, etc have no effect cannot be made without evidence that adequate KD of the protein was achieved. The western blots showing adequate knockdown need to be presented.

Minor issues:

Supp Fig 2e - the control band in the western is labeled b-actin but the legend says a-tubulin was

used as the control.

Line 134-5: "ablating Prmt5 had no obvious impact on the JDP2/PRMT5 association". Presumably the authors meant the JDP2/ β -catenin interaction.

Please check the key to the graph in Fig. 7b. Presumably the JDP2 results are presented by the yellow bar and vector by the dark red bar. If so, please amend the key or the color of the bars.

Reviewer #3 (Remarks to the Author):

Synopsis: In this manuscript by Cao et al., the authors present data in support of a TCF-independent transcriptional role for β -catenin in response to genotoxic stresses including camptothecin (CPT), irradiation (IR) or cisplatin (CDDP). Most of the studies are undertaken in cell lines including 293FT (human embryonic kidney), OVCAR3 (human ovarian cancer), MCF-7 (human breast cancer) and A549 (human lung carcinoma), although the authors also provide data from an in vivo xenograft model using primary ovarian cancer tissues and conduct online Kaplan-Meier analysis of data from patients with ovarian, lung gastric or breast cancer, which suggest that high JDP2 levels correlate with cancer relapse and poorer patient outcome.

By conducting RNA-seq and ChIP-seq (for β -catenin) studies in CPT-treated 293FT cells, the authors observed an enrichment for genes with GO terms related to glutathione biosynthesis/metabolism, and they focused on three genes encoding key factors of GSH metabolism: *SLC7A1*, *GCLM* and *GSS*, which were bound by β -catenin and dependent on β -catenin for their regulation.

Immunoprecipitation of β -catenin-bound chromatin, followed by mass spectrometry of β -catenin co-associated proteins, identified JDP2 as a factor that coordinates with β -catenin to regulate target genes involved in GSH metabolism induced by CPT.

By using a series of biochemical and imaging approaches, the authors assemble a mechanism in which β -catenin competes with JDP2's binding to histone H3/H4 and promotes its binding to target DNA. Further assays conducted by the authors suggest that ATM kinase phosphorylates JDP2 upon genotoxic stress and promotes the association of JDP2 and the arginine methyltransferase PRMT5 to alter H3 methylation in the promoter of the target gene *SLC7A1* to promote transcription via recruitment of WDR5/MLL complexes.

Experiments in cell lines and mouse xenografts strongly suggest that JDP2 contributes to the resistance of cancer cells to genotoxic treatments. Finally, data are presented that suggest that co-treatment with genotoxic agents and inhibitors of WDR/MLL may act synergistically in anti tumour treatments.

Critical Overview:

This study represents a huge amount of work, and overall, I find the data presented to be convincing and intriguing. The experiments have been undertaken with appropriate controls and multiple lines of evidence support many of the findings. I believe the study will be of great interest to a broad audience, as it provides new insights into transcriptional roles for β -catenin that are uncoupled from conventional Wnt/ β -catenin signalling. The data are compelling and raise as many questions as they answer, which heightened my enthusiasm for the paper. I have only one major concern concerning the TCF-independence of β -catenin's transcriptional role, which should be readily addressable by the authors.

Major Concerns:

The authors rule out a role for TCF/LEFs primarily based on the use of RNAi-mediated knockdown of TCF4. Most cell types, including the cell lines used in this study express more than one of the four TCF/LEF factors, so knocking down a single factor does not definitively rule out a role for TCF/LEFs in the effects attributed to β -catenin. Recently, cell lines lacking all four TCF/LEF factors have been generated and described (Moreira et al. Cell Rep. 2017 Sep 5;20(10):2424-2438. and Doumpas et al., EMBO J. 2019 Jan 15;38(2).) To conclusively rule out a role for TCF/LEFs in their mechanism for β -catenin-mediated target gene regulation, the authors should conduct key experiments in cells lacking TCF/LEFs (e.g. key experiments in which knockdown of TCF4 was used to rule out a role for TCF/LEFs).

Minor Concerns:

Scientifically relevant typographical errors:

Line 53 "intercellular" should be "intracellular"

Line 391 "campathecin" should be "camptothecin"

2. The authors have not cited a key paper in which a functional interaction between β -catenin and FOXO was shown in oxidative stress signalling (Essers, et al., Science. 2005 May 20;308(5725):1181-4.). As the data in supplemental figure 1 use Foxo3 as a positive control for a β -catenin interactor in conditions of genotoxic stress, this paper helps the reader understand why Foxo is a suitable control.

Editor's comments:

We therefore invite you to revise and resubmit your manuscript, taking into account the points raised. Notably, we require all relevant technical information and siRNA controls be to be provided (Reviewer #2) and for key experiments to be performed in cells lacking all four TCF/LEF transcription factor (Reviewer #3). In addition, more discussion on how GSH might be generating the observed outcomes with more detail on specific ROS species is necessary (Reviewer #1). Please highlight all changes in the manuscript text file.

Response: We do appreciate the editor for highlighting these important points raised by the reviewers, to which we respond as the following:

(1) As requested by Reviewer #2, more detailed technical information regarding ChIP-seq analyses and IP-MS studies, as well as the western blotting results of siRNA controls, have been provided in the Results and Materials and Methods section in the revised manuscript (Please see responses to reviewers below for details).

(2) As requested by Reviewer #3, the key experiments in the cells lacking all four TCF/LEF transcription factor were performed and relevant results have been incorporated into the revised manuscript (Please see responses to reviewers below for details and new Supplementary Fig. 3a-3c).

(3) As requested by Reviewer #1, more detailed discussion on more detailed discussion on how GSH generated the observed outcomes and more detail on specific ROS species have been added in Introduction and Discussion section in the revised manuscript (Please see responses to reviewers below for details).

(4) Meanwhile, as requested by the editor, all changes have been highlighted in our newly revised manuscript text file.

Reviewers' comments:

Reviewer #1 (Remarks to the Author):

This is a densely written paper with focus on factors that contribute to regulation of gene clusters relevant to certain component parts of redox homeostasis. In general terms, the English narrative is patchily presented and should be reconsidered by someone with English as a first language. Redox homeostasis is discussed in generic and sometimes vague terms and the term ROS (reactive oxygen species) is used throughout as if it is one entity. The experiments appear to be reliably presented and their interpretation appears reasonable throughout. The biggest concern is the fact that there are few mechanistic considerations as to how ROS mediated chemical reactions are transmitted through the transcriptional regulatory pathways studies to produce the end results reported. For example, the authors do not distinguish hydrogen peroxide, hydroxyl radical, superoxide (or some other species?) or how the oxidative stress stimulates downstream events. As presented, the interference with protein:protein interactions and subsequent effects on protein complexes presumably occurs through ROS mediated changes in protein structures. Is this post-translational modifications (in this case maybe S-glutathionylation?), protein stability or something different? Camptothecin, cisplatin and radiation will produce quite distinct reactive chemical species and these will be dose dependent. Also, cytotoxicities of these agent may not all be directly linked to GSH.

Response: We do appreciate the reviewer's comments and thank the reviewer for raising these important points. As requested by the reviewer, we have carefully edited the whole manuscript, added more detailed information of the terms of "Redox homeostasis" and "ROS (reactive oxygen species)" in Introduction and Discussion section, and added the necessary data in the revised manuscript, to which we respond as the following:

(1) As suggested by the reviewer, we have carefully edited the entire manuscript and the manuscript has been polished by a professional editor before resubmission.

(2) As requested by the reviewer, more detailed information of "Redox homeostasis" and "Reactive oxygen species (ROS)" has been added and discussed in Introduction and Discussion section in the revised manuscript. Reactive oxygen species (ROS) are composed of free radicals with unpaired electron and non-radical oxygen species containing oxygen such as superoxide ($O_2^{\cdot-}$), hydrogen peroxide (H_2O_2), singlet oxygen (1O_2) and the hydroxyl radical ($HO\cdot$) (1-3). Acting as signaling molecules, ROS are essential for the efficient and proper execution of a large number of cellular processes, such as the regulation of

intracellular signal transduction and gene expression patterns (1-3). However, excessive or prolonged ROS generation results in considerable damage to cellular constituents, various diseased conditions and the process of ageing (4-5). On the other hand, intracellular thiols, such as glutathione (GSH), cysteine (Cys) and homocysteine (Hcy) play a crucial role in defense against oxidative stress and scavenging of ROS, resulted in maintaining biological redox homeostasis (6-7). As the most abundant endogenous low molecular weight redox molecule within mammalian cells, glutathione (GSH) plays pleiotropic roles in preventing damage induced by either external or intracellular stimuli. GSH could either function as an antioxidant to directly scavenge ROS or serve as an electron donor for other redox systems, such as glutaredoxin (Grx) and glutathione peroxidase (GPX), to scavenge peroxide-related products (8-10). The abovementioned descriptions have been added in Introduction section in the revised manuscript.

Reduced Glutathione (GSH), functions as both a nucleophile and a reductant, plays roles in various cellular processes through regulation of the thiol-redox status. GSH could effectively scavenges multiple types of ROS (e.g., hydroxyl radical, lipid peroxy radical, superoxide anion, and hydrogen peroxide) via non-enzymatic reduction or eliminate hydroperoxides required enzymatic catalysis (1-3). Hence, maintaining or reestablishing intracellular GSH homeostasis is fundamental for cellular physiological functions, such as cell survival and tissue regeneration. Several mechanisms have been reported whereby cells maintain their GSH redox state in response to oxidative stress, such as *de novo* synthesis and salvage pathways (8-10). The abovementioned descriptions have been added in Discussion section in the revised manuscript.

(3) We thank for the reviewer's comments and the specific types of ROS induced by camptothecin (CPT), or Cisplatin (CDDP), or ionizing radiation have been discussed in the revised manuscript. Previously, Sen et al. have reported that CPT, an inhibitor of DNA topoisomerase I, induced ROS elevation, especially the level of superoxide radical and hydroxyl radical, which eventually promoted apoptotic cell death via dysfunction of cellular respiration and mitochondrial hyperpolarization (11). Marullo et al found that CDDP treatment upregulated ROS levels in cancer cells was dependent on mitochondria instead of nuclear DNA damage signaling in DU145 and DU145 ρ° cells (lacking mitochondrial DNA) (12). Consistently, Tarpey MM et al demonstrated that hydroxyl radical scavenger was essential for ameliorating the nephrotoxicity following CDDP chemotherapy via protecting mitochondria and preventing oxidative stress (13). Consistently, exposing cells to either γ -

radiation or α -particles significantly enhanced cellular ROS levels, such as superoxide anions and hydrogen peroxide, via inducing a reversible mitochondrial permeability transition above that attributable to normal cell metabolism (14-15). Therefore, genotoxic stress induced by inhibitors of DNA topoisomerase, or chemotherapeutic drug (CDDP), or γ -radiation could induce multiple types of ROS. Meanwhile, multiple research groups have documented that depletion of GSH could potentiate apoptosis provoked by CPT, or CDDP, or ionizing radiation, suggesting that cytotoxicity provoked by these reagents are linked to intracellular GSH alteration (11, 16-17). The abovementioned descriptions have been added in Discussion section in the revised manuscript.

(4) We do appreciate the reviewer's comments and the reviewer raised very an important question. Redox-dependent post-translational modification (PTM), such as S-glutathionylation of sulfur-containing amino acids, controls a wide range of intracellular protein activities and is involved in the response of oxidative stress (18-19). For instance, it has been reported that S-glutathionylation promoted TAZ stability that was critical for ROS-mediated transactivation of TAZ (20), and oxidative stress-induced S-glutathionylation of mitochondrial thymidine kinase 2 (TK2) has significant impacts on degradation of TK2 and mitochondrial DNA precursor synthesis (21). In the current study, we demonstrated that β -catenin formed a complex with ATM phosphorylated-JDP2 and PRMT5 that associated with the promoters of multiple genes in the GSH-metabolic cascade, resulted in reestablishing GSH homeostasis upon genotoxic stress. As suggested by the reviewer, we further examined whether genotoxic stress-induced S-glutathionylation and stability of JDP2, β -catenin, ATM and PRMT5. As shown in Supplementary Figure 7a-7b, genotoxic stress did not induce either S-glutathionylational modification or stabilization of JDP2, β -catenin, ATM and PRMT5, which provided further evidence that genotoxic stress-triggered β -catenin/JDP2/PRMT5 complex facilitated reestablishing glutathione homeostasis via *de novo* GSH synthesis. The abovementioned results and descriptions have been added in the Result and Discussion section in the revised manuscript.

References

- (1) Ray PD, et al. Reactive oxygen species (ROS) homeostasis and redox regulation in cellular signaling. *Cell Signal*. 2012; 24:981-990.
- (2) Locato V, et al. ROS and redox balance as multifaceted players of cross-tolerance: epigenetic and retrograde control of gene expression. *J Exp Bot*. 2018; 69:3373-3391.
- (3) Shadel GS, et al. Mitochondrial ROS signaling in organismal homeostasis. *Cell*. 2015; 163:560-569.

- (4) Waris G, et al. Reactive oxygen species: role in the development of cancer and various chronic conditions. *J Carcinog.* 2006; 5:14.
- (5) Yu BP. Cellular defenses against damage from reactive oxygen species. *Physiol Rev.* 1994; 74:139-162.
- (6) Ursini F, et al. Redox homeostasis: The Golden Mean of healthy living. *Redox Biol.* 2016; 8:205-15.
- (7) Willems PH, et al. Redox homeostasis and mitochondrial dynamics. *Cell Metab.* 2015; 22:207-18.
- (8) Pompella A, et al. The changing faces of glutathione, a cellular protagonist. *Biochem Pharmacol.* 2003; 66:1499-503.
- (9) Aquilano K, et al. Glutathione: new roles in redox signaling for an old antioxidant. *Front Pharmacol.* 2014; 5:196.
- (10) Fujii J, et al. Unveiling the roles of the glutathione redox system in vivo by analyzing genetically modified mice. *J Clin Biochem Nutr.* 2011; 49:70-8.
- (11) Sen N, et al. Camptothecin induced mitochondrial dysfunction leading to programmed cell death in unicellular hemoflagellate *Leishmania donovani*. *Cell Death Differ.* 2004; 11:924-936.
- (12) Marullo R, et al. Cisplatin induces a mitochondrial-ROS response that contributes to cytotoxicity depending on mitochondrial redox status and bioenergetic functions. *PLoS one.* 2013; 8, e81162.
- (13) Santos NA, et al. Hydroxyl radical scavenger ameliorates cisplatin-induced nephrotoxicity by preventing oxidative stress, redox state unbalance, impairment of energetic metabolism and apoptosis in rat kidney mitochondria. *Cancer Chemother Pharmacol.* 2008; 61:145-155.
- (14) Leach JK, et al. Ionizing radiation-induced, mitochondria-dependent generation of reactive oxygen/nitrogen. *Cancer Res.* 2001; 61:3894-3901.
- (15) Narayanan P., et al. α particles initiate biological production of superoxide anions and hydrogen peroxide. *Cancer Res.* 1997; 57: 3963-3971.
- (16) Ma MZ, et al. Xc⁻ inhibitor sulfasalazine sensitizes colorectal cancer to cisplatin by a GSH-dependent mechanism. *Cancer Lett.* 2015; 368:88-96.
- (17) Jayakumar S, et al. Differential response of DU145 and PC3 prostate cancer cells to ionizing radiation: role of reactive oxygen species, GSH and Nrf2 in radiosensitivity. *Biochim Biophys Acta.* 2014; 1840:485-94.
- (18) Kim HJ, et al. ROSics: chemistry and proteomics of cysteine modifications in redox biology. *Mass Spectrom Rev.* 2015; 34:184-208.
- (19) Toren Finkel. Signal transduction by reactive oxygen species. *J Cell Biol.* 2011; 194:7-15.
- (20) Gandhirajan RK, et al. Cysteine S-Glutathionylation Promotes Stability and Activation of the Hippo Downstream Effector Transcriptional Co-activator with PDZ-binding Motif (TAZ). *J Biol Chem.* 2016; 291:11596-607.
- (21) Sun R, et al. Oxidative stress induced S-glutathionylation and proteolytic degradation of

mitochondrial thymidine kinase 2. J Biol Chem. 2012; 287:24304-12.

Minor issues

1. While the primary focus of this paper is *de novo* GSH synthesis, GSH homeostasis can also be influenced by salvage pathways, for example through γ -GGT. There is little indication that this is considered.

Response: The reviewer raised a very important point and we thank the reviewer for this comment. Indeed, as mentioned by the reviewer, the salvage pathways, such as γ -Glutamyl transferase (GGT) and thioredoxin/glutaredoxin (TRX/GRX), have been also demonstrated to contribute to GSH homeostasis. For instance, γ -GGT could enhance cellular GSH synthesis by increasing the availability of component amino acids, and TRX/GRX regulate cellular GSH homeostasis by reduction of oxidized forms, such as glutathione disulfide (GSSG) and glutathione mixed disulfide with protein thiols (GS-R), back to the reduced form of GSH (1-3). However, our RNA-seq analysis data showed that mRNA expression of γ -GGT, TRX and GRX was not significantly changed, but levels of *SLC7A11*, *GCLM*, and *GSS* gene were significantly increased, in the genotoxic stress-treated cells, suggesting that genotoxic stress-activated β -catenin signaling facilitated the restoration of GSH metabolism via *de novo* GSH synthesis. The abovementioned descriptions and results have been added in Introduction and Result section in the revised manuscript.

References

- (1) Whitfield JB. Gamma glutamyl transferase. Crit Rev Clin Lab Sci. 2001; 38:263-355.
- (2) Kondo N, et al. Redox regulation of human thioredoxin network. Antioxid Redox Signal 2006; 8:1881-1890.
- (3) Meyer Y, et al. Thioredoxins and glutaredoxins: unifying elements in redox biology. Annu Rev Genet 2009; 43: 335-367.

2. It is interesting that SLC7A11 is expressed at such high levels in the tumor cell lines used. Normally, significant expression is associated with brain and neurons.

Response: We thank for the reviewer's comment. As mentioned by the reviewer, SLC7A11, also named as xCT, is normally highly expressing in brain and neurons. Interestingly, several research groups have recently reported that expression of SLC7A11 is significantly upregulated in multiple human cancer types, such as non-small cell lung cancer (NSCLC), liver carcinoma and melanoma, which contributes to progression and development of cancer via regulation of redox homeostasis and metabolic reprogramming (1-4).

References

- (1) Ji X, et al. xCT (SLC7A11)-mediated metabolic reprogramming promotes non-small cell lung cancer progression. *Oncogene*. 2018; 37:5007-5019.
- (2) Zhang L, et al. Overexpression of SLC7A11: a novel oncogene and an indicator of unfavorable prognosis for liver carcinoma. *Future Oncol*. 2018; 14:927-936.
- (3) Wang L, et al. An Acquired Vulnerability of Drug-Resistant Melanoma with Therapeutic Potential. *Cell*. 2018; 173:1413-1425.
- (4) Koppula P, et al. Amino acid transporter SLC7A11/xCT at the crossroads of regulating redox homeostasis and nutrient dependency of cancer. *Cancer Commun (Lond)*. 2018; 38:12

3. Lines 66/67 do not make sense....”Interestingly, the genotoxic stress-mediated reduction of GSH level could speedily recovery back and even further elevated after a few hours.”

Response: We thank for the reviewer’s comment and we are sorry that we did not write this point clearly in the originally submitted manuscript. Previously, Zegura B et al reported that exposure of HepG2 cells to microcystin-LR (MCLR) , which could induce DNA strand breaks, led to a significant reduction (by 35% compared with control cells) of intracellular GSH content after 10 min. During the next 2.0 h, the content of GSH began to increase gradually and was still below the content in control cells, but it was close to the level in control cells after 4.0 h of exposure. However, the maximal level of GSH in MCLR treated cells was observed, which was around 1.7 times than that in control cells, after 6.0 h of exposure (Fig. 2) (1). Meanwhile, Ghibelli L, et al. found that exposure of U973 cells to topoisomerase II inhibitor etoposide (VP-16) resulted in a reduction of intracellular GSH content at 3.0 h (85% vs. 100% in control cells), but after 4.5 h, the level of GSH in VP-16-treated cells was increased to 110% compared to control cells (Table 1) (2). Consistent with the results in these previous reports (1-2), we also observed that the relative GSH level quickly reduced in the CPT-treated cells within 10 mins but recovery back to un-treated level around 4.0 h and further elevated up to 2.5-fold after 6.0 h of treatment (Figure 1i). Integration of previous reports and our results showed in the current study suggested that genotoxic stress-mediated reduction of GSH level could speedily recovery back and even further elevated after a few hours.

References

- (1) Zegura B, et al. Alteration of intracellular GSH levels and its role in microcystin-LR-induced DNA damage in human hepatoma HepG2 cells. *Mutat Res*. 2006; 611: 25-33.
- (2) Ghibelli L, et al. Non-oxidative loss of glutathione in apoptosis via GSH extrusion. *Biochem Biophys Res Commun*. 1995; 216:313-20.

4. Line 279.....it is not clear what is meant by “biological free radicals.” This also raises the issue that there is little discussion of which type of electrophilic species or which nucleophilic (amino acid?) targets are critical to the observed transcriptional events.

Response: We thank the reviewer for this comment and we are sorry that we used a vague term “biological free radicals” in the manuscript, which has been deleted in the revised manuscript. We have re-discussed the GSH hemostasis in Discussion section in the revision. Reduced Glutathione (GSH), functions as both a nucleophile and a reductant, plays roles in various cellular processes through regulation of the thiol-redox status. GSH could effectively scavenge multiple types of ROS (e.g., hydroxyl radical, lipid peroxy radical, superoxide anion, and hydrogen peroxide) via non-enzymatic reduction or eliminate hydroperoxides required enzymatic catalysis (1-3). In the current study, we demonstrated that in response to genotoxic stresses induced by camptothecin (CPT), or Cisplatin (CDDP), or γ -radiation, β -catenin formed a complex with ATM phosphorylated-JDP2 and PRMT5 that associated with the promoters of in the multiple GSH-metabolic genes, resulted in reestablishing GSH homeostasis. Interestingly, it has been previously reported that CPT could induce ROS elevation, especially the level of superoxide radical and hydroxyl radical (4), and hydroxyl radical was significantly increased in the cells following CDDP (5), and exposing cells to γ -radiation significantly enhanced cellular ROS levels, such as superoxide anions and hydrogen peroxide (6). Hence, genotoxic stresses provoked by CPT, or CDDP, or γ -radiation could induce elevation of multiple types of ROS. However, multiple research groups have also documented that depletion of GSH could potentiate apoptosis provoked by CPT, or CDDP, or ionizing radiation, suggesting that cytotoxicity provoked by these reagents are linked to intracellular GSH alteration (4, 7-8). Recently, redox-dependent post-translational modification have been recently reported to be involved in the response of oxidative stress, such as S-glutathionylation of sulfur-containing amino acids (9-10). It has been reported that S-glutathionylation promoted TAZ stability that was critical for reactive oxygen species (ROS)-mediated transactivation of TAZ, and oxidative stress-induced S-glutathionylation of TK2 has significant impacts on TK2 degradation and mitochondrial DNA precursor synthesis (11-12). In the current study, we demonstrated that β -catenin formed a complex with phosphorylated-JDP2 and PRMT5 that associated with the promoters of multiple genes in the GSH-metabolic cascade, resulted in reestablishing GSH homeostasis upon genotoxic stress. As suggested by the reviewer, we further examined whether genotoxic stress-induced S-glutathionylation and stability of JDP2, β -catenin, ATM and PRMT5. As shown in Supplementary Figure 7a-7b, genotoxic stress did not induce either S-glutathionylational

modification or stabilization of JDP2, β -catenin, ATM and PRMT5, which provided further evidence that genotoxic stress-triggered β -catenin/JDP2/PRMT5 complex facilitated reestablishing GSH homeostasis via *de novo* GSH synthesis. The abovementioned results and descriptions have been added in the Result and Discussion section in the revised manuscript.

References

- (1) Pompella A, et al. The changing faces of glutathione, a cellular protagonist. *Biochem Pharmacol.* 2003; 66:1499-503.
- (2) Aquilano K, et al. Glutathione: new roles in redox signaling for an old antioxidant. *Front Pharmacol.* 2014; 5:196.
- (3) Fujii J, et al. Unveiling the roles of the glutathione redox system in vivo by analyzing genetically modified mice. *J Clin Biochem Nutr.* 2011; 49(2):70-8.
- (4) Sen N, et al. Camptothecin induced mitochondrial dysfunction leading to programmed cell death in unicellular donovani. *Cell Death Differ.* 2004; 11:924-936.
- (5) Santos NA, et al. Hydroxyl radical scavenger ameliorates cisplatin-induced nephrotoxicity by preventing oxidative stress, impairment of energetic metabolism and apoptosis in rat kidney mitochondria. *Cancer Chemother Pharmacol.* 2008; 61:145-155.
- (6) Leach JK, et al. Ionizing radiation-induced, mitochondria-dependent generation of reactive oxygen/nitrogen. *Cancer research* 2001; 61:3894-3901.
- (7) Ma MZ, et al. Xc- inhibitor sulfasalazine sensitizes colorectal cancer to cisplatin by a GSH-dependent mechanism. *Cancer Lett.* 2015; 368:88-96.
- (8) Jayakumar S, et al. Differential response of DU145 and PC3 prostate cancer cells to ionizing radiation: role of reactive oxygen species, GSH and Nrf2 in radiosensitivity. *Biochim Biophys Acta.* 2014; 1840:485-94.
- (9) Kim HJ, et al. ROSics: chemistry and proteomics of cysteine modifications in redox biology. *Mass Spectrom Rev.* 2015; 34:184-208.
- (10) Toren Finkel. Signal transduction by reactive oxygen species. *J Cell Biol.* 2011;194:7-15.
- (11) Gandhirajan RK, et al. Cysteine S-Glutathionylation promotes stability and activation of the Hippo downstream effector transcriptional co-activator with PDZ-binding Motif (TAZ). *J Biol Chem.* 2016; 291:11596-607.
- (12) Sun R, et al. Oxidative stress induced S-glutathionylation and proteolytic degradation of mitochondrial thymidine kinase 2. *J Biol Chem.* 2012; 287:24304-12.

5. Line 286...."cellular GSH levels would be speedily exhausted and the cells become unable to counteract the ROS-mediated insults, which would lead to irreversible cell degeneration and death." This is a kinetic analysis without data.

Response: We thank for the reviewer's comment and reviewer's point is well taken.

Although multiple studies have documented that when cells exposure to genotoxic stress, GSH level was significantly reduced, ROS level and apoptotic cells were significantly

increased (1-3), none of these studies simultaneously conducted kinetic analysis to show the correlation among GSH levels, ROS level and apoptotic cells. We have added more detailed information of “Redox homeostasis” and “reactive oxygen species (ROS)” in the revised manuscript, and this sentence has been deleted in the revised manuscript.

References

- (1) Ghibelli L, et al. Non-oxidative loss of glutathione in apoptosis via GSH extrusion. *Biochem Biophys Res Commun.* 1995; 216:313-20.
- (2) Zegura B, et al. Alteration of intracellular GSH levels and its role in microcystin-LR-induced DNA damage in human hepatoma HepG2 cells. *Mutat Res.* 2006; 611:25-33.
- (3) Sen N, et al. Camptothecin induced mitochondrial dysfunction leading to programmed cell death in unicellular hemoflagellate *Leishmania donovani*. *Cell Death Differ.* 2004; 11:924-936.

6. Line 311.....”Genotoxic stress, such as DNA-damaging agents, topoisomerase inhibitors, and cisplatin, could sharply induce DNA damage, higher ROS levels, and stronger cytotoxicity compared with that produced by H₂O₂ treatment.” Again, this does not really make sense, but presumably the implication is that DNA damage is more relevant to cytotoxicity than hydrogen peroxide. How is this linked to drug induced changes in GSH homeostasis?

Response: We do appreciate the reviewer for the comments and the reviewer`s points are well taken. We have added more detailed information regarding the genotoxic stress provoked by DNA-damaging agents, topoisomerase inhibitors and cisplatin, and have deleted these imprecise descriptions in the revised manuscript. Previously, it has been reported that β -catenin interacts with and induce FOXO-mediated transcription, resulting in removal of H₂O₂ via upregulation of manganese superoxide dismutase and catalase, which was through a TCF-independent mechanism (1-3). Meanwhile, genotoxic stress-induced poly-ADP-ribosylated PARP-1 and upregulated Ku70, which competed for the interaction of TCF4 with β -catenin and reduced β -catenin/TCF transcriptional activity (4-5). However, the biological role of β -catenin signaling in response to genotoxic stress remains unclear. In the present study, we observed that, unlike H₂O₂ treatment, genotoxic stresses did not increase the β -catenin/FOXOs interaction, suggesting that the effect of β -catenin signaling on the reduction of genotoxic stress-induced ROS might be through other mechanisms. We further demonstrated that genotoxic stress induced the rapid enrichment of β -catenin in chromatin, where it formed a complex with ATM phosphorylated-JDP2 and PRMT5, which facilitated the restoration of GSH homeostasis via transcriptional upregulation of multiple genes in the

GSH-metabolic cascade, including SLC7A11, GCLM, and GSS, resulting in elimination of genotoxic stress-induced ROS. Therefore, our results revealed a novel mechanism by which β -catenin signaling maintains redox homeostasis in genotoxic stress-treated cells.

References

- (1) Almeida M, et al. Oxidative stress antagonizes Wnt signaling in osteoblast precursors by diverting beta-catenin from T cell factor- to forkhead box O-mediated transcription. *J Biol Chem.* 2007; 282:27298-27305.
- (2) Hoogeboom D, et al. Interaction of FOXO with beta-catenin inhibits β -catenin/T cell factor activity. *J Biol Chem.* 2008; 283: 9224-9230.
- (3) Shin SY, et al. Involvement of glycogen synthase kinase-3beta in hydrogen peroxide-induced suppression of Tcf/Lef-dependent transcriptional activity. *Cell Signal.* 2006; 18:601-607.
- (4) Idogawa M, et al. Ku70 and poly(ADP-ribose) polymerase-1 competitively regulate β -catenin and TCF4-mediated gene transactivation: possible linkage of DNA damage recognition and Wnt signaling. *Cancer Res.* 2007; 67:911-918.
- (5) Tavana O, et al. Ku70 functions in addition to nonhomologous end joining in pancreatic beta-cells: a connection to beta-catenin regulation. *Diabetes* 2013; 62: 2429-2438.

7. Line 343. “However, emerging evidence demonstrated that GSH confers protection against chemoradiotherapy in cancer by reducing genotoxic stress-induced ROS (refs 9, 40, 41).”

This statement oversimplifies the depth of existing literature for the field of GSH pathways and drug resistance, and presents two papers on AsO₃ and one that does not seem relevant at all. In classifying ROS as a composite of all chemical species, there is little to distinguish between radiation and all chemotherapeutic agents.

Response: We thank the reviewer for the comment and the reviewer’s point is well taken. As suggested by the reviewer, we have rewritten the Discussion section in the revised manuscript, which introduced and distinguished the specific types of ROS induced by chemotherapeutic drug, such as CPT and CDDP, and ionizing radiation. And we also added the detailed information regarding the role of GSH in chemodrug resistance and radiotherapy resistance. Meanwhile, the appropriate references (1-2) have been added, which replaced the previous references, in the revised manuscript.

References

- (1) Traverso N, et al. Role of glutathione in cancer progression and chemoresistance. *Oxid Med Cell Longev.* 2013; 2013:972913.
- (2) Bansal A, Simon MC. Glutathione metabolism in cancer progression and treatment resistance. *J Cell Biol.* 2018; 217(7):2291-2298.

Reviewer #2 (Remarks to the Author):

Cao et al have submitted an interesting paper defining a β -catenin/JDP2/PRMT5 axis in mediating the cellular response to genotoxic stress. The paper is wonderfully detailed in its analysis of mechanism, and the mechanistic insights are interesting and novel. The authors provide correlation data linking JDP2 and poor patient outcome for a number of human cancer parameters. Finally, they demonstrate functional outcomes of manipulating JDP2 in multiple tumor models to demonstrate that targeting JDP2 increases the chemosensitivity of genotoxic chemotherapeutic agents and therefore supports the idea that this approach could be considered as an adjuvant therapy for current chemotherapeutic regimens. **I believe the authors have supported their conclusions thoroughly, and will not ask for additional experiments.**

My primary concern is that technical standards for data presentation and analysis are not met.

1. The authors do not include relevant information on the ChIP-seq analysis they performed. Only by reading the Reporting summary can one learn that the ChIP-seq data are from a single replicate with 23 million reads. That might be an adequate number of reads, but it is unclear if that is total reads or mapped reads, or something else. I would think that at minimum there should be a duplicate sample and there should be regression analysis showing the extent of correlation between the datasets. Was input sequenced? Fig legend 1F says peak density was analyzed, but the Reporting Summary makes reference to MACS for peak calling. I find no reference to data obtained via MACS in the manuscript. Inputting the RNA-seq and the ChIP-seq accession numbers for the GEO datasets generated a “not found” message. The ChIP-seq data set was analyzed for Go terms and for location relative to genes. A very small percentage of β -catenin binding sites were in “promoters”, and the authors presented binding relative to TSS sites of genes, but the authors do not indicate how many peaks were identified across the genome. Was it 100 or 10000? Did those peaks correlate with any known cis-element sequences? Are there other data sets for these cells in the literature, perhaps indicating areas of histone modification, or anything else, that could be examined for overlap?

Response: We do appreciate the reviewer for these comments and all the reviewer's points are well taken, and we are also sorry that we did not write these points clearly in our originally submitted manuscript. Our response is as follows:

(1) The detailed information of ChIP-seq dataset analysis have been added in Result and

Materials and Methods section in the revised manuscript. Meanwhile, as requested by the reviewer, one more duplicate genotoxic stress-treated sample was examined by ChIP-seq assay. Therefore, the information of ChIP-seq dataset analysis was the results obtained from 2 samples, which regression analysis showed significant the extent of correlation between these datasets ($r = 0.85$; $P < 1.0 \times 10^{-10}$ by Spearson's chi-squared test). The input of both samples were sequenced in the current study. We do appreciate the reviewer for pointing out the error. It should be "peak calling" in the Figure legend 1F, and appropriate correction has been made in the revised manuscript. Analysis of ChIP-seq data showed that there are total 23 million reads and 19727 peaks were identified across the genome (raw data accessible via PRJNA543097) in genotoxic stress-treated 293FT cells. The above mentioned results and descriptions have been added in the revised manuscript. Interestingly, we found that, in addition to cis-element sequence of JDP2 transcriptional factor, the cis-element sequence of multiple transcriptional factors, such as Sp1, Sp2 and PLAG1, were also correlated with these peaks. However, our mass spectrometry (MS) did not show that these factors were interacted with β -catenin in CPT-treated cells, which suggested that transcriptional factor Sp1, Sp2 and PLAG1 might not be involved in genotoxic stress-triggered β -catenin signalling induced GSH homeostasis. Meanwhile, we did not find any no histone modification datasets for the genotoxic stress-treated these cells in the previously published literature.

(2) The detailed information about the MACS method for ChIP-seq assay has been added in Materials and Methods section in the revised manuscript. The reads were first mapped the human genome sequence (hg19) by using bowtie2 (version 2.2.9). The parameters were set as default, SAMtools (version 0.1.19) was then used to convert files to bam format, sort. Peaks were called using MACS (version 2.1.1) (1-3), with $FDR \leq 0.05$. A peak was assigned to the transcriptional start site (TSS) of a RefSeq gene when falling into the surrounding 4kb (± 2 kb). Promoters were defined as 6kb regions (± 3 kb) surrounding the TSS. The promoter peaks analysis, KEGG Pathway and GO analysis was implemented using KOBAS (version 3.0). *P*-value was calculated by student's t-test.

(3) We are sorry that we wrote wrong number using submission ID in the originally submitted manuscript, which should be the BioProject ID. We have corrected these errors in the revised manuscript, which ChIP-seq and RNA-seq datasets have been deposited in National Center for Biotechnology Information (NCBI, <http://www.ncbi.nlm.nih.gov/>) with accession code PRJNA543097 and PRJNA543096. We are willing to release these datasets after our manuscript being accepted. However, we would like to release these datasets if the

reviewer prefers.

References

- (1) Zhang Y, et al. Model-based analysis of ChIP-Seq (MACS). *Genome Biol.* 2008; 9: R137.
- (2) Feng J, et al. Identifying ChIP-seq enrichment using MACS. *Nature protocols.* 2012; 7, 1728-1740.
- (3) Yevshin I, et al. GTRD: a database of transcription factor binding sites identified by ChIP-seq experiments. *Nucleic acids research.* 2017; 45, D61-D67.

2. The authors performed MS following IP with β -catenin antibody to identify interacting proteins. They should mention in the text of the Results that this was done using cells transfected with a β -catenin expression vector so that it is immediately understood that the experiment was performed under overexpression conditions. A gel with some of the identified bands is presented along with a representative MS plot to show the sequences identified as JDP2 and PRMT5. There is no discussion of the MS analysis - just a statement that it was performed. No experimental details are provided whatsoever. These should be presented in detail so that those who are interested understand the settings and cutoffs that were used can. It is unlikely that the results were limited to the proteins identified in the gel. Shouldn't there be a comprehensive list of the proteins identified, and preferably, the peptide sequences identified by MS, the number of hits per protein, the coverage achieved? That is a pretty basic requirement for IP-MS studies.

Response: We thank the reviewer for raising these important points and we are also sorry that we did not write these points clearly in the originally submitted manuscript. Our response is as follows:

(1) As suggested by the reviewer, the detailed information about the experimental process and the information about the cells that was transduced with a β -catenin expression vector have been added in the Materials and Methods and Results section in the revised manuscript.

(2) The detailed information of IP-MS studies have been added in Result section in the revised manuscript, and as requested by the reviewer, the list of the proteins identified by MS, including the proteins and the peptide sequences identified by MS, the number of hits per protein, the coverage achieved, have been also summarized as Supplementary Tables 3 and 4 in the revised manuscript.

3. In Figure 2b and 2c, and the accompanying Supplementary figures, the authors show that

siRNA mediated knockdown against all of the candidate interacting proteins do or do not affect expression, factor binding, etc. In the case where effects are seen, they follow-up and present the westerns showing the effectiveness of the siRNA. However, the negative conclusions stating that SMARCA4, PARP, TCF4, etc have no effect cannot be made without evidence that adequate KD of the protein was achieved. The western blots showing adequate knockdown need to be presented.

Response: We thank for the reviewer`s comment and the reviewer`s points is well taken. As requested by the reviewer, we further performed western blotting analysis of expression of SMARCA4, JDP2, HNRNPA2B1, FOXO3, TCF4, PRMT5, β -catenin, α -catenin, PARP1 to examine the knockdown effectiveness of respective siRNA. As shown in Supplementary Fig. 4a, the expression of abovementioned protein was dramatically decreased in the siRNA-transfected cells. The abovementioned results have been added into the revised manuscript.

Minor issues:

1. Supp Fig 2e – the control band in the western is labeled β -actin but the legend says a-tubulin was used as the control.

Response: We apologize for the typing error in the Figure legend of Supp Fig 2e and thank the reviewer for pointing it out. It should be “ β -actin” and appropriate correction has been made in the revised manuscript.

2. Line 134-5: “ablating Prmt5 had no obvious impact on the JDP2/PRMT5 association”. Presumably the authors meant the JDP2/ β -catenin interaction.

Response: We do appreciate the comment and thank the reviewer for pointing out the error. It should be “ablating PRMT5 had no obvious impact on the JDP2/ β -catenin association”, and appropriate correction has been made in the revised manuscript.

3. Please check the key to the graph in Fig. 7b. Presumably the JDP2 results are presented by the yellow bar and vector by the dark red bar. If so, please amend the key or the color of the bars.

Response: We apologize for the error in Fig. 7b and thank the reviewer for pointing it out. Appropriate correction has been made in the revised manuscript.

Reviewer #3 (Remarks to the Author):

Critical Overview:

This study represents a huge amount of work, and overall, I find the data presented to be convincing and intriguing. The experiments have been undertaken with appropriate controls and multiple lines of evidence support many of the findings. I believe the study will be of great interest to a broad audience, as it provides new insights into transcriptional roles for β -catenin that are uncoupled from conventional Wnt/ β -catenin signalling. The data are compelling and raise as many questions as they answer, which heightened my enthusiasm for the paper. **I have only one major concern concerning the TCF-independence of β -catenin's transcriptional role, which should be readily addressable by the authors.**

Major Concerns:

1. The authors rule out a role for TCF/LEFs primarily based on the use of RNAi-mediated knockdown of TCF4. Most cell types, including the cell lines used in this study express more than one of the four TCF/LEF factors, so knocking down a single factor does not definitively rule out a role for TCF/LEFs in the effects attributed to β -catenin. Recently, cell lines lacking all four TCF/LEF factors have been generated and described (Moreira et al. Cell Rep. 2017 Sep 5; 20(10):2424-2438. and Doumpas et al., EMBO J. 2019 Jan 15; 38(2).) To conclusively rule out a role for TCF/LEFs in their mechanism for β -catenin-mediated target gene regulation, the authors should conduct key experiments in cells lacking TCF/LEFs (e.g. key experiments in which knockdown of TCF4 was used to rule out a role for TCF/LEFs).

Response: We do appreciate the reviewer's comment and the reviewer raised a very important question. As suggested by the reviewer, in order to conclusively rule out a role for TCF/LEFs in the genotoxic stress activated- β -catenin signaling, we further individually silenced other three TCF/LEF factors, including TCF1, LEF1 and TCF3, and examined the effects attributed to β -catenin signalling in the genotoxic stress-treated cells (Supplementary Fig. 3a). As shown in Supplementary Figure 3b-3c, consistent with the effect of knockdown of TCF4, silencing of either TCF1, or LEF1, or TCF3 in genotoxic agent-treated cells did not alter the expression of SLC7A1, GCLM and GSS and had no effect on the enrichment of β -catenin on the promoters of these GSH-metabolic genes. These results further supported the notion that genotoxic stress activated- β -catenin signaling was through a TCF/LEF-independent mechanism. The abovementioned results and descriptions have been added in the revised manuscript (please see new Supplementary Fig. 3a-3c).

Minor Concerns: Scientifically relevant typographical errors:

1. Line 53 “intercellular” should be “intracellular”

Response: We apologize for the typing error and thank the reviewer for pointing it out. Appropriate correction has been made in the revised manuscript.

2. Line 391 “campathecin” should be “camptothecin”

Response: We apologize for the typing error and thank the reviewer for pointing this out. Appropriate correction has been made in the revised manuscript.

3. The authors have not cited a key paper in which a functional interaction between β -catenin and FOXO was shown in oxidative stress signalling (Essers, et al., Science. 2005 May 20; 308(5725):1181-4.). As the data in Supplementary figure 1 use Foxo3 as a positive control for a β -catenin interactor in conditions of genotoxic stress, this paper helps the reader understand why Foxo is a suitable control.

Response: We do appreciate reviewer’s comment and the reviewer’s point is well taken. As suggested by the reviewer, the previous important work regarding the functional interaction between β -catenin and FOXO in oxidative stress signalling (Essers, et al., Functional interaction between beta-catenin and FOXO in oxidative stress signaling. Science. 2005; 308(5725):1181-4.) has been cited in the revised manuscript.

REVIEWERS' COMMENTS:

Reviewer #1 (Remarks to the Author):

It is still unclear what type of "ROS" is causing the noted effects or what links the cause:effect consequences to the signaling events.

They do provide additional negative data for post-translational changes and expression alterations (RNA seq) in salvage of GSH.

I note that the other two reviewers appear to be significantly more enthusiastic about the paper than me. As such, and in recognizing the significant existing bulk of data, and that I suspect the authors will not provide the chemical data I would have no major concern should you move forward with accepting the paper.

Reviewer #2 (Remarks to the Author):

I am sorry, but the authors' response to one query is unclear. This has to do with scientific rigor and with data transparency and presentation for large datasets, not with the suitability of the manuscript for publication.

In version 1, the authors reported their ChIP-seq analysis involved 23 million reads from a single replicate. In the response letter that accompanies the revised manuscript, they indicated they have performed a second replicate with results that significantly correlated with those of the first replicate. Yet the revised text says that "Analysis of the b-catenin ChIP-seq data from duplicate samples with total 23 million reads...".

Shouldn't the number of reads have increased if a second replicate was performed? Or did each replicate have 23 million reads? Or was a replicate performed and analyzed for correlation coefficient, but the data from this second replicate were not analyzed independently or as part of a pooled dataset? In other words, the analyses that were reported were based only on the replicate 1 dataset?

As I type this out, I'm reasonably convinced the third possibility is the answer. If I'm correct, this is disappointing. Why not pool the data and re-determine the number and location of peaks? It's not like there are multiple or complicated analyses that are based on the ChIP-seq that would be difficult to repeat.

Ideally,

(1) the ChIP-seq data should be pooled and the peak analysis redone, reporting the number of reads, the number of peaks, and a complete accounting of the number or percentage of peaks that are promoter associated, intergenic, intragenic, intronic, etc.

(2) I previously commented that the authors did not indicate whether the 23 million reads reported were total or mapped. They respond that they are total, but then they did not indicate the number of mapped reads. This is a standard thing to do, for each replicate and for the input, often in the form of a short table incorporated into the supplemental data, though the format is not important.

(3) The authors included in the response letter that they had evaluated the correlation between the replicates, but they do share this information in the manuscript. The information provided in the response letter should be incorporated into the manuscript methods section or supplemental data.

If the authors cannot pool the data and re-analyze, they should explicitly indicate that the data they are reporting are based on one of the two replicates and points (2) and (3) above still need to be

addressed.

Reviewer #3 (Remarks to the Author):

Although the authors did not obtain truly TCF/LEF deficient cells lacking all 4 family members to test the TCF/LEF-independence of their β -catenin-mediated effects, I am pleased that they at least examined the other TCF/LEF factors with their knockdown approach, although this approach does not rule out functional compensation by the TCF/LEF factors that remain after individual factors are knocked down. Indeed, their new data reveal that all 4 TCF/LEF factors are expressed at detectable protein levels in their system, so the possibility of functional redundancy of the various TCF/LEF factors is a valid concern regarding the interpretation of results obtained when only one of the factors has its expression reduced. If the authors mention this caveat in their discussion, I will be satisfied.

Editor's comments:

We therefore invite you to revise your paper one last time to address the remaining concerns of our reviewers. Notably, you must address the comments of Reviewer #2 regarding the number of total vs mapped reads, pool the data using both replicate data sets, and provide correlation of the two datasets in the manuscript. You must also include the caveat regarding potential TCF/LEF functional redundancy mentioned by Reviewer #3 in your discussion. Both of these are required for acceptance of your revised manuscript.

At the same time we ask that you edit your manuscript to comply with our format requirements and to maximise the accessibility and therefore the impact of your work.

Response: We do appreciate the editor for highlighting these important points,

(1) As suggested by Reviewer #2, we re-analyzed the pooled ChIP-seq data using both replicate datasets and provided more detailed analysis results, including the number of total and mapped reads and the significant correlation of two datasets, in the revised manuscript.

(2) As suggested by Reviewer #3, the caveat regarding potential TCF/LEF functional redundancy have been discussed in Discussion section in the revised manuscript.

(3) As requested by the editor, we have carefully gone through the manuscript and edited our manuscript to comply with journal policies and formatting style.

Reviewers' comments:**Reviewer #1 (Remarks to the Author):**

It is still unclear what type of "ROS" is causing the noted effects or what links the cause: effect consequences to the signaling events. They do provide additional negative data for post-translational changes and expression alterations (RNA seq) in salvage of GSH. I note that the other two reviewers appear to be significantly more enthusiastic about the paper than me.

As such, and in recognizing the significant existing bulk of data, and that I suspect the authors will not provide the chemical data **I would have no major concern should you move forward with accepting the paper.**

Response: We deeply thank the reviewer for his (her) appreciation on our tremendous efforts in addressing all the concerns.

Reviewer #2 (Remarks to the Author):

In version 1, the authors reported their ChIP-seq analysis involved 23 million reads from a

single replicate. In the response letter that accompanies the revised manuscript, they indicated they have performed a second replicate with results that significantly correlated with those of the first replicate. Yet the revised text says that “Analysis of the β -catenin ChIP-seq data from duplicate samples with total 23 million reads...”. Shouldn't the number of reads have increased if a second replicate was performed? Or did each replicate have 23 million reads? Or was a replicate performed and analyzed for correlation coefficient, but the data from this second replicate were not analyzed independently or as part of a pooled dataset? In other words, the analyses that were reported were based only on the replicate 1 dataset?

As I type this out, I'm reasonably convinced the third possibility is the answer. If I'm correct, this is disappointing. Why not pool the data and re-determine the number and location of peaks? It's not like there are multiple or complicated analyses that are based on the ChIP-seq that would be difficult to repeat.

Ideally, (1) the ChIP-seq data should be pooled and the peak analysis redone, reporting the number of reads, the number of peaks, and a complete accounting of the number or percentage of peaks that are promoter associated, intergenic, intragenic, intronic, etc.

(2) I previously commented that the authors did not indicate whether the 23 million reads reported were total or mapped. They respond that they are total, but then they did not indicate the number of mapped reads. This is a standard thing to do, for each replicate and for the input, often in the form of a short table incorporated into the supplemental data, though the format is not important.

(3) The authors included in the response letter that they had evaluated the correlation between the replicates, but they do share this information in the manuscript. The information provided in the response letter should be incorporated into the manuscript methods section or supplemental data.

If the authors cannot pool the data and re-analyze, they should explicitly indicate that the data they are reporting are based on one of the two replicates and points (2) and (3) above still need to be addressed.

Response: We do appreciate the reviewer's comments. As suggested by the reviewer, we re-analyzed the pooled ChIP-seq data using both replicate data sets. The detailed analysis of the results have been summarized in the Supplementary Table 1 and also added in Results section in the revised manuscript. Statistical analysis revealed that there were 57.4 million total reads and 55.6 million mapped reads, and 20521 peaks were identified, including 6925 peaks in promoter, 4265 peaks in intergenic, 6821 peaks in intron, 1112 peaks in exon, 688

peaks in 5`UTR, 125 peaks in 3`UTR, 365 peaks in TTS (Please see Supplementary Table 1). Meanwhile, the significant correlation between two ChIP-seq datasets ($P < 1.0 \times 10^{-10}$, $r = 0.85$ by Spearson`s chi-squared test) has been added in Results section and in Supplementary Table 1 in the revised manuscript.

Reviewer #3 (Remarks to the Author):

Although the authors did not obtain truly TCF/LEF deficient cells lacking all 4 family members to test the TCF/LEF-independence of their β -catenin-mediated effects, I am pleased that they at least examined the other TCF/LEF factors with their knockdown approach, although this approach does not rule out functional compensation by the TCF/LEF factors that remain after individual factors are knocked down. Indeed, their new data reveal that all 4 TCF/LEF factors are expressed at detectable protein levels in their system, so the possibility of functional redundancy of the various TCF/LEF factors is a valid concern regarding the interpretation of results obtained when only one of the factors has its expression reduced. **If the authors mention this caveat in their discussion, I will be satisfied.**

Response: We thank the reviewer for the positive comment on our revised manuscript. As suggested by the reviewer, the caveat regarding potential TCF/LEF functional redundancy have been discussed in Discussion section in the revised manuscript.